# Analysis of the Spatial and Temporal Variation of Sea Ice and Connectivity in the NEP of the Arctic in Summer in Hot Years

Guochong Liu [1] , Min Ji [1,]*, Fengxiang Jin [1], Ying Li [1], Yawen He [2] and Ting Li [1]

1   Surveying and Spatial Information Institute, Shandong University of Science and Technology,
    Qingdao 266590, China; lgc1172843080@163.com (G.L.); fxjin@sdjzu.edu.cn (F.J.); kdsgliying@163.com (Y.L.);
    liting_sdust@126.com (T.L.)
2   Ocean and Spatial Information College, China University of Petroleum, Qingdao 266000, China;
    heyw@upc.edu.cn
*   Correspondence: jamesjimin@126.com; Tel.: +86-178-0626-3036

**Abstract:** Climate warming has enabled the Arctic region to achieve seasonal navigation, and sea ice concentration is an important factor affecting the navigation of the Arctic waterways. This article uses the Arctic sea ice concentration data of the three highest temperatures in 2016, 2019, and 2020, combined with the Arctic summer sea level pressure, wind field, temperature, temperature anomaly, ice age, and sea ice movement data to analyze the spatial and temporal variation of sea ice and connectivity in the Northeast Passage (NEP) of the Arctic in Summer in three hot years, and summarizes the causes of sea ice anomalies. The results show that: (1) the summer Arctic sea ice extent in 2016, 2019 and 2020 were all lower than the multi-year average sea ice extent, and the summer sea ice extent in 2020 had the largest change trend; (2) in October of these three years, the sea ice was all negative anomalies, extending the opening time of the NEP; (3) when the sea ice concentration was 30% as the threshold, the navigation period of the NEP in 2016 was from mid-August to late October, 2019 was from the beginning of August to mid-October, 2020 was from the end of July to the end of October, and 2020 was the longest year since the opening of the NEP; (4) when the sea ice concentration was 10% as the threshold, the navigation period of the NEP in 2016 was from the end of August to the end of October, 2019 was from early August to mid-October, and 2020 was from the beginning of August to the end of October; (5) the key navigable areas of the NEP in the past three years were the central waters of the East Siberian Sea, the New Siberian Islands and the Vilkitsky Strait; (6) the navigation period of the NEP in 2016, 2019 and 2020 was longer. The main reasons were that the temperature of the NEP in the past 3 years was relatively high, the wind was weak, the sea ice movement had little effect, and the sea ice age in the key navigable areas was first year ice, which was easy to melt, which greatly promoted the opening of the NEP.

**Keywords:** Arctic; the NEP; hot year; navigation period; sea ice

## 1. Introduction

Since satellite observations in the 1970s, Arctic sea ice has been showing a decreasing trend. In the past 30 years, the rate of sea ice reduction has accelerated significantly [1–3], and the thickness of sea ice has been continuously thinning [4,5]. With the gradual warming of the global climate and the repeated ablation of Arctic sea ice, the Northeast Passage (NEP) had gradually achieved seasonal navigation since 2013, specifically after 2015, the opening period of the NEP reached 80 days. According to the National Aeronautics and Space Administration (NASA) [6], as shown in Figure 1. Since 1980, the annual temperature has shown a gradual upward trend. The global average surface temperature in 2020 was equal to that of 2016, which was the hottest year on record. Compared with the average temperature in 1951–1980, the average temperature in 2016 and 2020 both increased by 1.02 °C over the multi-year average, and in 2019 it increased by 0.99 °C. Due to the high

temperature, 2020 was tied with 2019 and 2016 as the top three high temperature years on record.

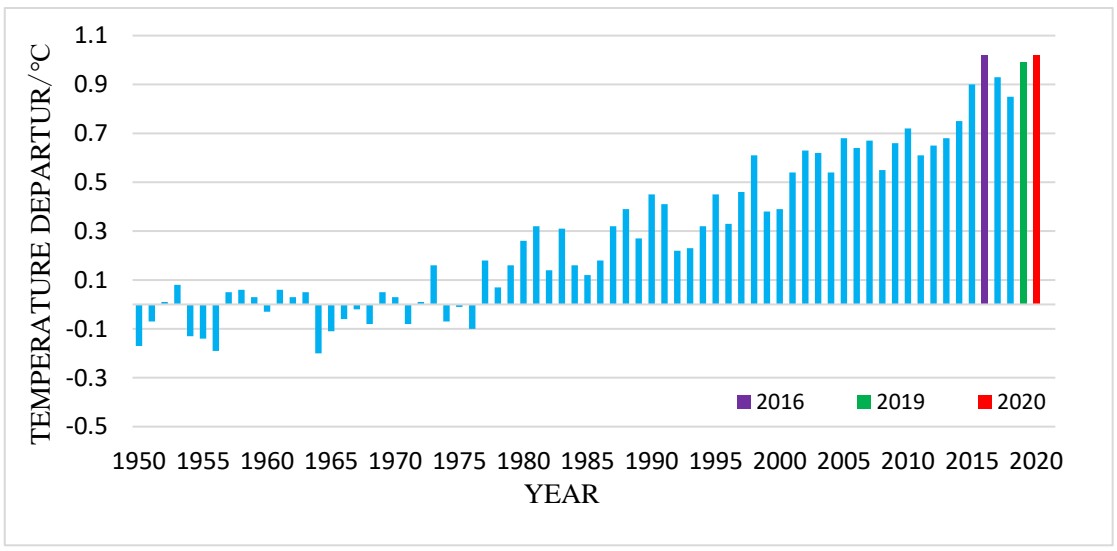

**Figure 1.** Histogram of global temperature anomalies (based on 1951–1980) from 1950 to 2020.

Rodrigues [7] analyzed the sea ice concentration data from 1979 to 2007 and found that Russia's Arctic sea ice coverage had been considerably reduced, and the length of the ice-free season had also increased. Cavalieri et al. [8] analyzed the changes in Arctic sea ice from 1979 to 2010 and found that the extent of Arctic sea ice decreased at a rate of $-4.1\% \pm 0.3\%$ every 10 years. Except for the Bering Sea, the extent and area of sea ice in other Arctic regions showed a clear decreasing trend. Smith et al. [9] used the Arctic September sea ice concentration and thickness data to analyze the climate prediction model of sea ice characteristics, thereby obtaining the Arctic optimal sailing route. Lei et al. [10] used remote sensing data from 1979 to 2012 to analyze the temporal and spatial changes of sea ice in the NEP, and found that the sea ice thickness of the NEP decreased by 0.7–1.0 m in the past 10 years, and the opening period of the NEP also increased. Dawei Gui et al. [11] used National Snow and Ice Data Center (NSIDC) products to analyze the kinematic characteristics of sea ice in the northern part of the NEP, and found that the sea ice moving northward in the Kara sector of the NEP in spring increased by 0.04 cm/s per year, and the sea ice moving northward improved the navigation of the NEP. Howell et al. [12] analyzed the impact of sea ice changes on shipping based on sea ice changes in the Northwest Passage (NWP) from 1969 to 2002. Zixuan Li et al. [13] analyzed and studied the changes in Arctic coastal terrestrial ice and found that Arctic terrestrial ice mainly appeared from January to June, distributed in the Canadian archipelago, the East Siberian Sea, the Laptev Sea and the Kara Sea. During 1976–2018, the change rate of the Arctic terrestrial ice extent was $-1.1 \pm 0.5 \times 10^4 \text{ km}^2/\text{year}$, and the change rate of the entire Arctic sea ice extent was $-6.0 \pm 2.4 \times 10^4 \text{ km}^2/\text{year}$.

Li Chunhua et al. [14] used microwave satellite remote sensing data to analyze the ice conditions and opening conditions of the NEP and NWP in the Arctic from 2002 to 2013, and found that the opening period of the NEP was mainly concentrated in late August to early October. The opening time was mostly 40–50 days, and the key area for opening of the channel was the Northland Islands Severnaya Zemlya. Xue Yanguang et al. [15] used the Arctic sea ice concentration data from 1972 to 2012 to calculate and analyze the changes in the extent of Arctic sea ice. The results showed that the Arctic sea ice had decreased significantly in the past 40 years, with the fastest decrease in September, and the decrease in Arctic sea ice lags behind the abnormally high temperatures in the northern hemisphere from February to April. Li Xinqing et al. [16] used sea ice concentration data from 2002 to 2013 to analyze the sea ice distribution characteristics and navigability of the Vilkitsky Strait,

and found that the navigable time of the strait was more than 40 days per year, and the opening time ranged from July to September, and the end time was relatively concentrated in October. Li Xinqing et al. [17] used AMSR-E and AMSR-2 sea ice concentration data from 2003 to 2014 from June to September to analyze the Arctic sea ice conditions, and found that the Arctic summer sea ice extent in 2014 was close to the 2003–2013 multi-year average. However, the sea ice in the Laptev Sea and its northern waters was significantly less than the multi-year median extent, while the northern archipelago of the Barents Sea was more than the multi-year median extent.

This article uses the Advanced Microwave Scanning Radiometer for EOS and the Advanced Microwave Scanning Radiometer 2(AMSR-E/AMSR2) Arctic sea ice concentration data from the recent 20 years of extremely high temperature, 2016, 2019 and 2020, combined with the Arctic summer sea level pressure, wind field, temperature, sea ice age, and sea ice motion data to analyze the spatio-temporal changes of sea ice and connectivity in the NEP in the summer of 2016, 2019 and 2020, and discussed the promoting factors of the navigable time of the NEP in these three years. The navigation problem of the NEP has always been of concern to all countries in the world. Exploring the causes of sea ice changes in the NEP and grasping the factors that affect the navigation period of the NEP has important reference significance for countries to use the NEP in a safer way.

## 2. Research Area, Data and Method

### 2.1. Research Area

The study area in this article is the NEP of the Arctic, which starts from the Bering Strait in the east, passing through the East Siberian Sea, the Laptev Sea, and the Kara Sea to the Barents Sea, with a total length of 5246.86 nautical miles. The NEP is divided into 5 segments. It is specifically divided as Table 1 and Figure 2.

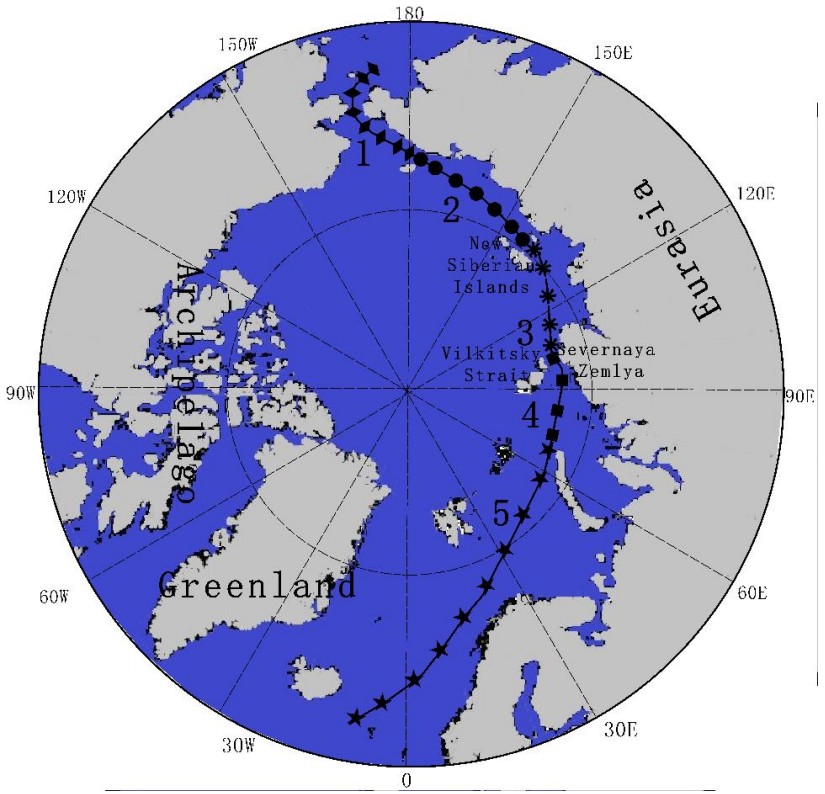

**Figure 2.** Schematic diagram of the Arctic Northeast Passage (NEP).

**Table 1.** Section division of the NEP.

| Section | Location |
| --- | --- |
| Bering Strait | 0–1311.71 nautical miles |
| East Siberian Sea | 1311.71–2098.74 nautical miles |
| Laptev Sea | 2098.74–2623.43 nautical miles |
| Kara Sea | 2623.43–3410.46 nautical miles |
| Barents Sea | 3410.46–3935.14 nautical miles |

According to the research of Ji, M et al. [18], the key areas of navigation in the NEP are mainly concentrated in the central part of the East Siberian Sea and the Vilkitsky Strait. The Vilkitsky Strait is the sea area with the most complicated and severe ice conditions in NEP and severely affects the opening of NEP.

### 2.2. Data

The sea ice concentration data come from the AMSR-E/AMSR2 high-resolution daily sea ice concentration data provided by the Institute of Environmental Physics (IUP) of the University of Bremen. The data set is AMSR2 Level 1B (https://seaice.uni-bre-men.de/start/, accessed on 5 May 2021) [19], Version 2.220.220. The spatial resolution of the data is 6.25 km, the time resolution is 1 day, and the time span is 2 July 2012 to today. The data are obtained by inversion of the ASI (Artist Sea Ice) algorithm [20], and the data have a high resolution, which can ensure the accuracy of the calculation of the navigation window of the NEP. Among them, the sea ice concentration data used in this article are from 1 January to 31 December in 2016, 2019 and 2020.

The sea ice extent data comes from the passive microwave data from the Defense Meteorological Satellite Program (DMSP) F17 and F18 Special Sensor Microwave Imager/Sound Device (SSMIS). The data set is Sea Ice Trends and Climatologies from SMMR and SSM/I-SSMIS (https://nsidc.org/data/NSIDC-0192/versions/3, accessed on 5 May 2021) [21], Version 3. The spatial resolution of the data is 25 km, and the time span is 1978–2020.

The sea level pressure, 2 m of near-surface air temperature and temperature anomalies come from NOAA's grid data set ERA5 (ESRL, https://psl.noaa.gov/repository/model/compare, accessed on 5 May 2021) [22], version 5. The time resolution is 1 month, the grid resolution is $0.25° \times 0.25°$, and the time span is from 1979 to 2021. Among them, the temperature anomaly is based on the average value from 2010 to 2019.

The wind field data come from the reanalyzed grid data set ERA5 (https://climatereanalyzer.org/reanalysis/monthly_maps/, accessed on 5 May 2021) of the University of Maine [23], version 5, it is a 10 m surface wind field. The time resolution is 1 month, the grid resolution is $0.5° \times 0.5°$, and the time span is from January 1950 to May 2021.

The sea ice age comes from EASE-Grid Sea Ice Age (NSIDC, https://nsidc.org/data/NSIDC-0611/versions/4, accessed on 5 May 2021) [24], version 4. The spatial resolution is 25 km $\times$ 25 km, the time resolution is 7 days, and the time span is 1 January 1984 to 31 December 2020.

The sea ice motion data come from the dataset of Polar Pathfinder Daily 25 km EASE-Grid Sea Ice Motion Vectors (https://nsidc.org/data/NSIDC-0116/versions/4, accessed on 5 May 2021) [25], version 4. The spatial resolution is 25 km $\times$ 25 km, the time resolution is 7 days, and the time span is from 25 October 1978 to 31 December 2020.

### 2.3. Method

#### 2.3.1. Calculation of Sea Ice Age and Sea Ice Movement

The netCDF data of sea ice age and sea ice movement in 2016, 2019 and 2020 are obtained from NSIDC. The time span is a whole year, and the time resolution is 1 week.

Using Matlab, the weekly data of sea ice age from 23 to 26 weeks are superimposed and averaged, and rounded up to the nearest integer, and the monthly ice age data in June are obtained. We overlay the weekly data of sea ice age from 27 to 30 weeks and take the average, and round up to obtain the monthly ice age data for July. We overlay the weekly

data of sea ice age from 31 to 35 weeks and take the average, and round up to obtain the monthly ice age data of August.

We use Matlab to read the U and V components of the sea ice motion data, and superimpose the U and V values of the sea ice motion data for 23 to 26 weeks, 27 to 30 weeks, and 31 to 35 weeks, and take the average values respectively to obtain the direction and speed of sea ice movement in June, July and August. When the U component is positive, it means that the sea ice is moving eastward. When the V component is positive, it means that the sea ice is moving northward. See the following formula (1) for details:

$$\vec{a} = \sqrt{u^2 + v^2} \tag{1}$$

### 2.3.2. Calculation of Sea Ice Anomalies

With reference to the calculation method of NSIDC (https://nsidc.org/arcticseaicene ws/sea-ice-analysis-tool/, accessed on 5 May 2021), the sea ice concentration of June, July and August from 2011 to 2020 are superimposed and averaged respectively to obtain the multi-year average for June, July and August. Then, the sea ice concentration in June, July, and August in the three years of 2016, 2019, and 2020 are calculated separately from the multi-year average to obtain the sea ice concentration anomaly in that month of the year.

### 2.3.3. Extraction of Sea Ice Concentration on the Northeast Passage (NEP)

We first draw the NEP within the Arctic, and extract the average sampling point coordinates on the drawn NEP. Using the latitude and longitude coordinates of the sea ice concentration product itself, we superimposed this on the same coordinate system with the NEP to obtain the original sea ice concentration data on the NEP. We performed interpolation analysis on the original data to generate a complete interpolation result. Then, the sampling point coordinates were used to extract the concentration value of this point from the interpolation result, and the sea ice concentration of the NEP was obtained in turn.

## 3. Analysis of the NEP Ice Condition and Connectivity in Hot Years

### 3.1. Sea Ice Extent Analysis

Figure 3 shows the extent of sea ice in the Arctic from 2011 to 2020. Among them, 2012 was the minimum year of the extent of sea ice in the Arctic on record, which was $3.4 \times 10^6$ km$^2$. Claire et al. [26] studied the extent of sea ice in the Arctic in 2012 and found that a strong storm in August 2012 caused the total ice layer in the Arctic to separate and melt, exposing the main sea ice group to wind and waves, which further promoted the melting of the sea ice group, and caused an anomaly in the extent of the sea ice in 2012. Although the sea ice extent in 2012 was the lowest value ever, the temperature in 2012 was only 0.65 °C higher than the average temperature in 1951–1980 (Figure 1), which was not an abnormally high temperature year. Hence, this article will not analyze 2012. Except for 2016, 2019 and 2020, the remaining 7 years of 2011–2018 are averaged for analysis.

It can be seen from Figure 3 that each year reached the minimum sea ice extent in mid-September. Except for 2012, the sea ice extent in September 2020 was the smallest, and the sea ice extent in 2016 and 2019 was similar. Except for the sea ice extent from January to March 2020, which was slightly higher than the multi-year average, the sea ice extent for 2016, 2019 and 2020 was lower than the multi-year average.

It can be seen from Figure 4 that from July to October, the monthly average sea ice extent of the three high-temperature years was smaller than the multi-year average. Compared with the multi-year average, from July to October 2016, the sea ice extent in October decreased the most, $4.9 \times 10^5$ km$^2$, followed by September, which decreased by $1.9 \times 10^5$ km$^2$. The difference between July and August was $1.1 \times 10^5$ km$^2$, $1.8 \times 10^5$ km$^2$. In July to October 2019, it was still October that the difference in sea ice extent was the largest, which was $9.1 \times 10^5$ km$^2$ less than the multi-year average, followed by August with $5.7 \times 10^5$ km$^2$. The difference between July and September was $4.2 \times 10^5$ km$^2$ and $3.1 \times 10^5$ km$^2$, respectively. The extent of sea ice reduction in 2020 was the most apparent.

Compared with the multi-year average, the sea ice extent in October 2020 was reduced by $1.36 \times 10^6$ km$^2$, and the sea ice extent in July, August, and September decreased by $6.0 \times 10^5$ km$^2$, $4.4 \times 10^5$ km$^2$, and $7.4 \times 10^5$ km$^2$, respectively.

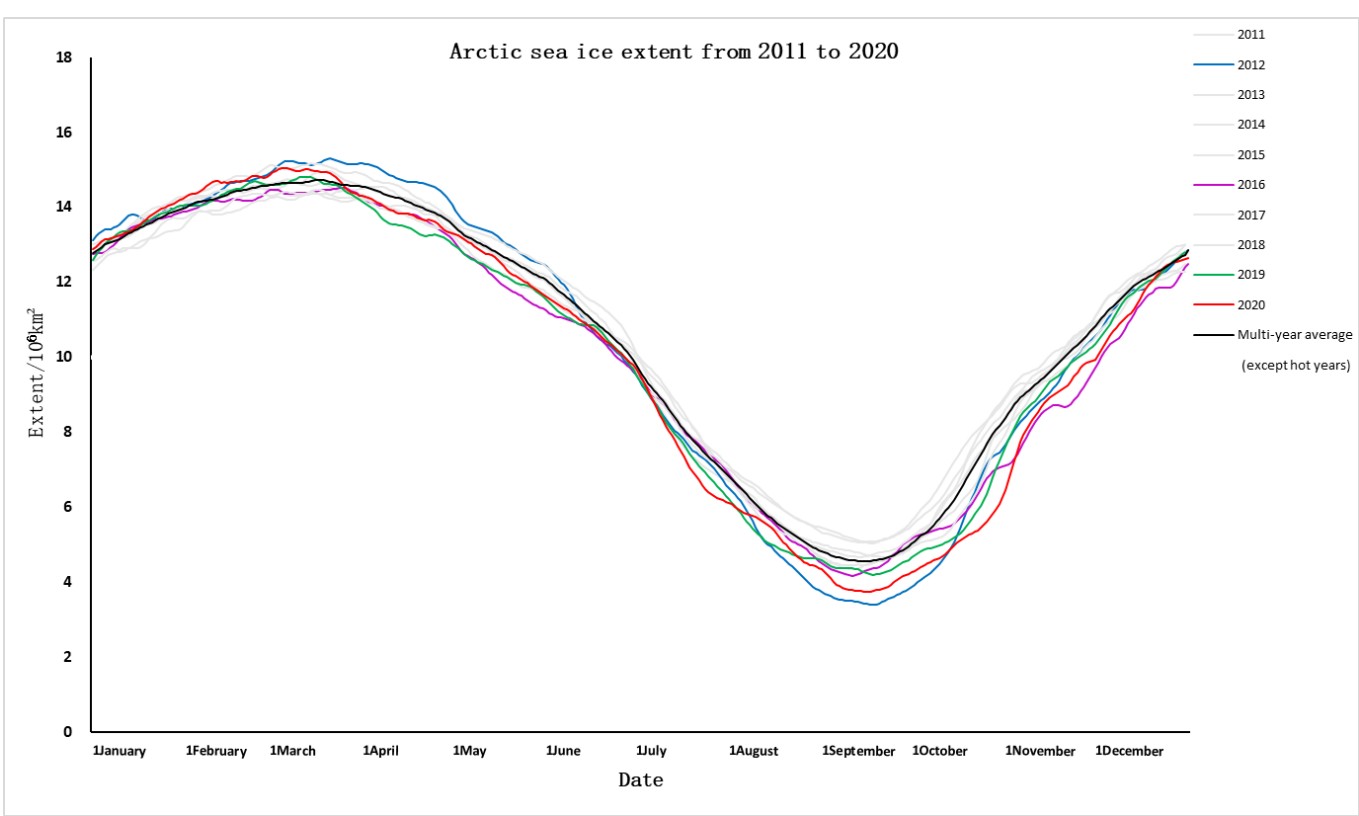

**Figure 3.** The extent of sea ice in the Arctic from 2011 to 2020.

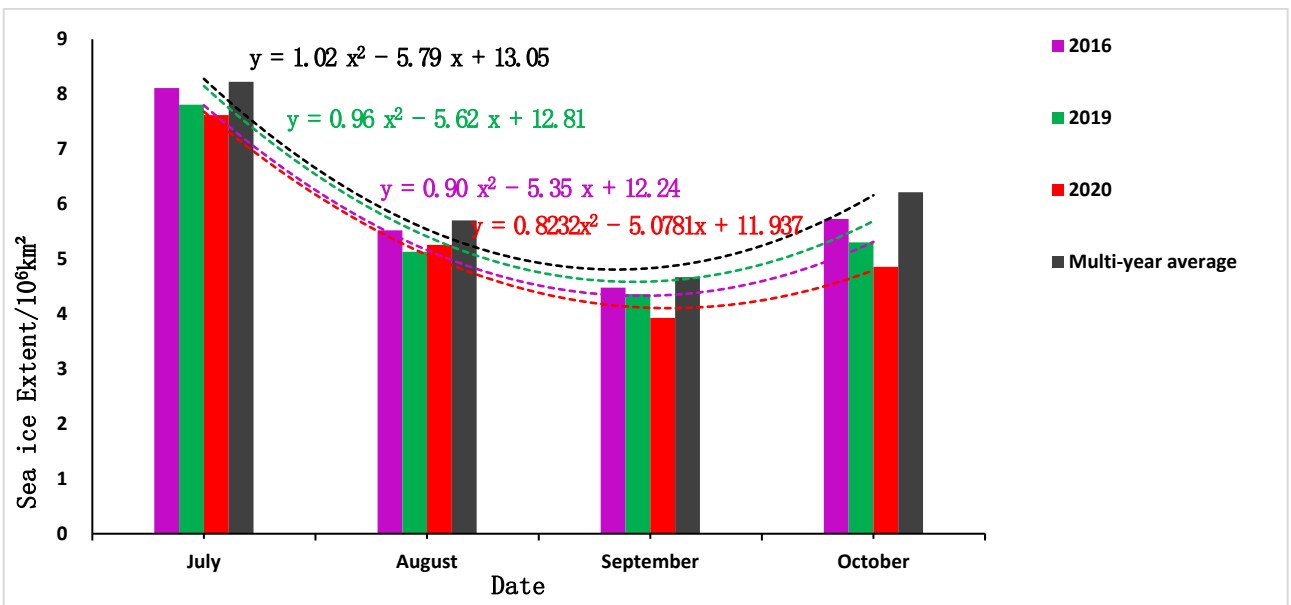

**Figure 4.** Histogram of monthly average sea ice extent in Arctic summer for 2016, 2019, 2020 and multi-year average. (The green dashed line is the fitting trend line of the sea ice extent in the summer of 2016, the purple dashed line is the summer of 2019, the red dashed line is the summer of 2020, and the black dashed line is the multi-year average).

From the fitting line of the three high-temperature years and the multi-year average in Figure 4, compared with the multi-year average, the extent of Arctic sea ice changes from July to October 2020 was the largest, followed by 2016, and the rate of change in 2019 was relatively tiny.

### 3.2. Sea Ice Concentration Analysis

In July 2016, the sea ice area in the Arctic was $5.3 \times 10^6$ km$^2$. It can be seen from Figure 5 that in the area covered by the NEP, about 60% of the sea ice was distributed in the East Siberian Sea and the Laptev Sea.

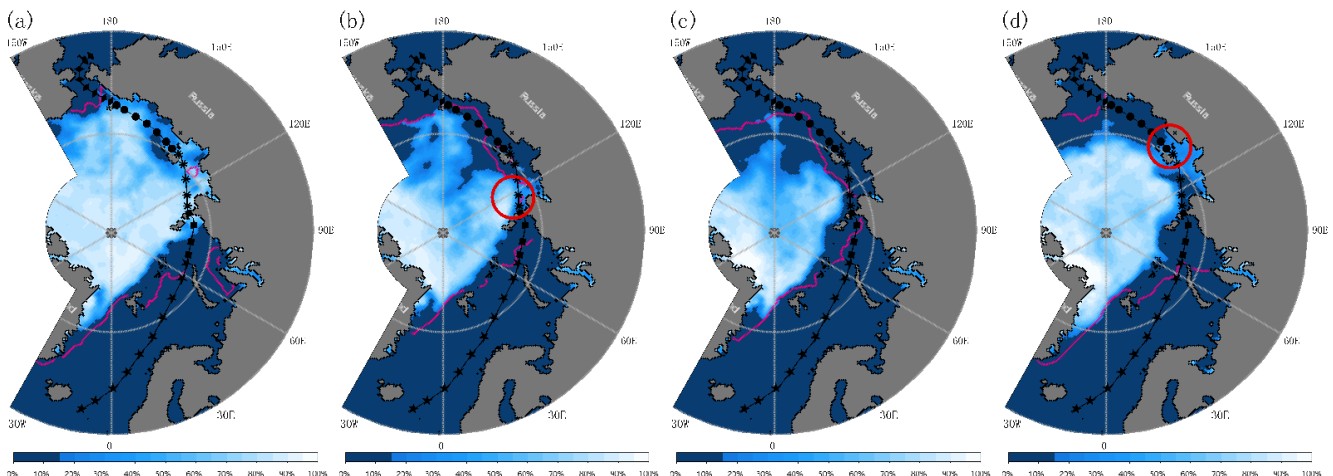

**Figure 5.** Spatial distribution map of sea ice concentration from July to October 2016 (**a–d**). (The red circles represent the special areas, the red line is the average sea ice edge line for each month from 1981 to 2010).

In August, the sea ice area was $3.3 \times 10^6$ km$^2$. In the NEP, only the area from the east of the Vilkitsky Strait to the Laptev Sea (red circle) had a small amount of 30% sea ice.

In September, the sea ice area reached the minimum value of $2.9 \times 10^6$ km$^2$. At this time, the sea ice of the NEP was nearly entirely melted, and large open ice-free sea areas appeared, and the NEP was unobstructed.

In October, the sea ice area in the Arctic was $4.3 \times 10^6$ km$^2$. The sea ice began to freeze and expanded to the vicinity of the NEP. About 20% of the sea ice was distributed near the New Siberian Islands (red circle).

Compared with the average sea ice edge line (red line) for each month from 1981 to 2010, the sea ice edge line from August to October 2016 was apparent, except for the small difference between the sea ice edge line in July 2016 and the multi-year average, the sea ice edge line from August to October 2016 was significantly smaller than the multi-year average.

In July 2019, the area of sea ice was $5.1 \times 10^6$ km$^2$. It can be seen from Figure 6 that most of the NEP was distributed with high-concentration sea ice. Both the Vilkitsky Strait and the East Siberian Sea were distributed with 30–60% of sea ice, making navigation hard.

The ice condition was good in August, and the sea ice area was $3.2 \times 10^6$ km$^2$. In the sea area passed by the NEP, only a small amount of sea ice was found on the east side of the Vilkitsky Strait.

In September, the sea ice area was $3.1 \times 10^6$ km$^2$, and the sea area covered by the NEP was ice-free, which was convenient for navigation.

In October, the Arctic sea ice entered a freezing period, and the sea ice concentration also increased. The sea ice area reached $4.5 \times 10^6$ km$^2$. In the NEP, 20–30% of sea ice was distributed on the east side of the Vilkitsky Strait.

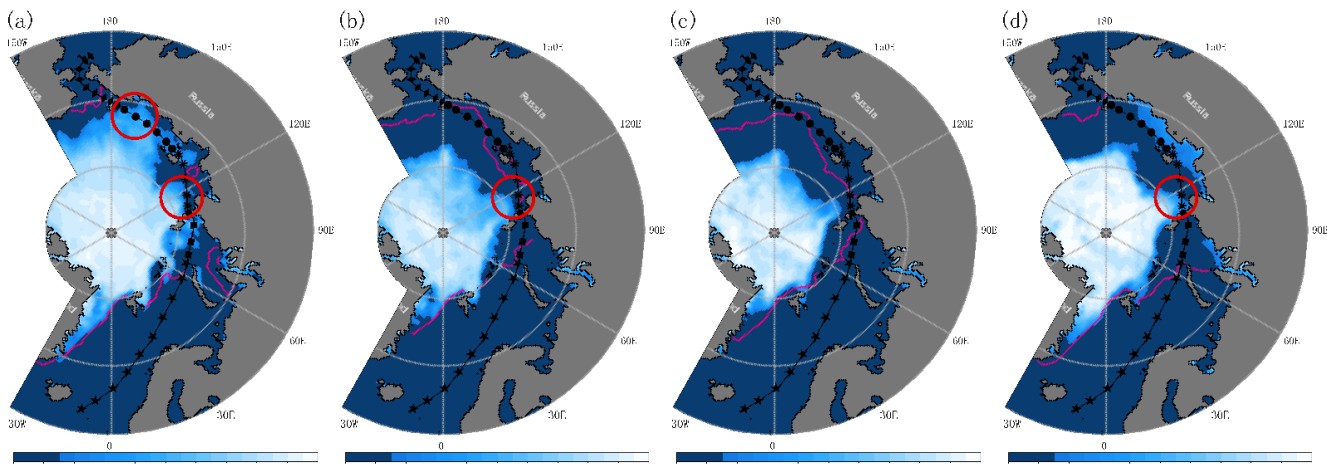

**Figure 6.** Spatial distribution map of sea ice concentration from July to October 2019 (**a–d**). (The red circles represent the special areas, the red line is the average sea ice edge line for each month from 1981 to 2010).

Compared with the average sea ice edge line (red line) for each month from 1981 to 2010, the sea ice edge line from July to October 2019 was still smaller than the multi-year average.

In July 2020, the sea ice area was $5.1 \times 10^6$ km². It can be seen from Figure 7 that the high concentration sea ice was mainly distributed in the central Arctic region, while near the NEP, the East Siberian Sea was distributed with a concentration of about 40% of sea ice, and a small amount of about 50% of sea ice was also distributed in the western part of the Vilkitsky Strait. The ice conditions in other sections were good.

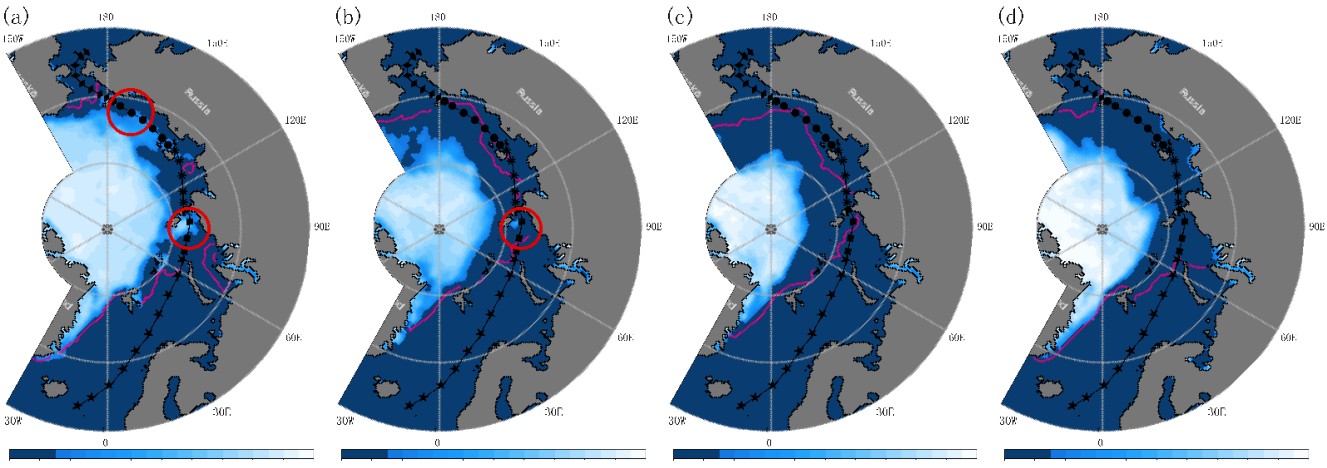

**Figure 7.** Spatial distribution map of sea ice concentration from July to October 2020 (**a–d**). (The red circles represent the special areas, the red line is the average sea ice edge line for each month from 1981 to 2010).

In August, the sea ice area was $3.1 \times 10^6$ km², sea ice concentrated in the central Arctic Ocean and the Canadian archipelago. In the NEP, only the western part of the Vilkitsky Strait had a tiny amount of sea ice with a concentration of less than 20%.

In September, the sea ice area was $2.8 \times 10^6$ km², sea ice continued to melt, the NEP had no sea ice distribution, and the entire route was unblocked.

Affected by climate lag, although the Arctic sea ice concentration increased slightly in October, the NEP was still in open water and the area of sea ice was $4.1 \times 10^6$ km².

Compared with the monthly average sea ice edge line (red line) from 1981 to 2010, the sea ice edge line from July to October 2020 was considerably smaller than the multi-year average.

### 3.3. Sea Ice Anomaly Analysis

It can be seen from Figure 8 that in July 2016, in the NEP, the sea ice concentration in the central area of the East Siberian Sea was higher than the average in previous years, and the sea ice concentration in the Laptev Sea was also higher than the multi-year average by more than 30%.

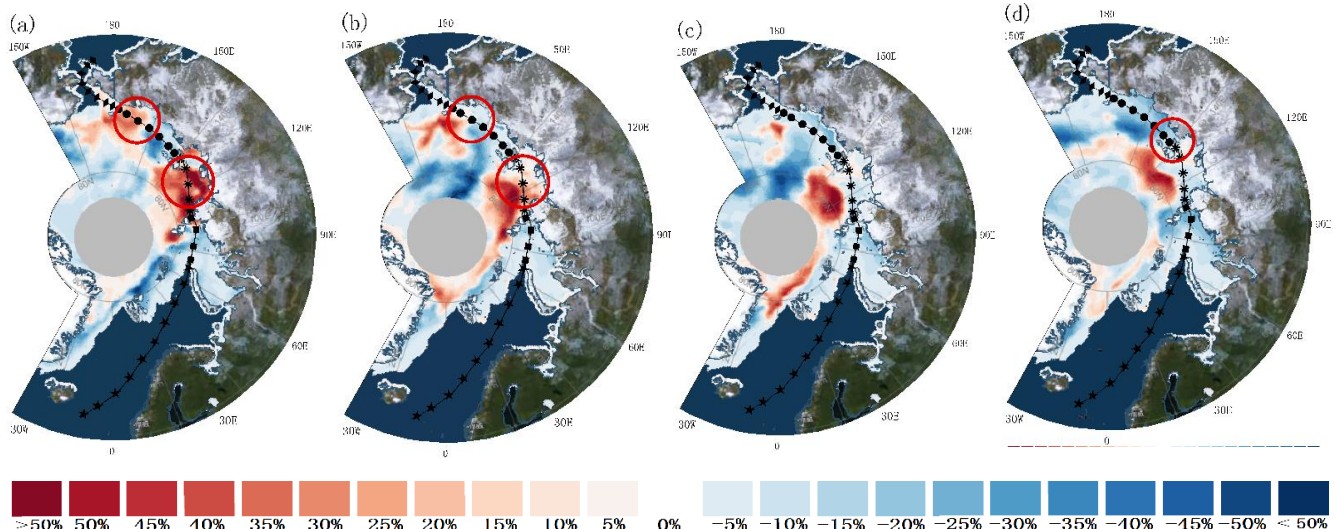

**Figure 8.** The anomalous distribution of sea ice concentration from July to October 2016 (**a–d**). (The outliers are based on the period from 2011 to 2020, the red circles represent the special areas).

In August, compared with the multi-year average, the sea ice concentration in the central East Siberian Sea and the Laptev Sea remained high.

In September, the sea ice concentration of the sea area through the NEP was not very different from previous years. It may be because 2011–2020 was the warmest 10 years in history. The NEP mostly had more ice-free sea areas in September, so the sea ice concentration of the NEP in September 2016 did not change significantly compared with the multi-year average.

In October, the sea ice concentration in the East Siberian Sea and the Vilkitsky Strait was low, while the sea ice in the vicinity of the New Siberian Islands increased.

It can be seen from Figure 9 that compared to the multi-year average, the sea ice concentration in most sea areas of the NEP was mainly reduced in July 2019. Except for the central part of the East Siberian Sea, the sea ice concentration was about 10% higher.

The decreasing trend of the NEP in August was not apparent.

In September, in the same way as in 2016, the NEP in September 2019 was basically ice-free water, and there was basically no change in sea ice each year.

In October, compared with the multi-year average, the sea ice concentration of the NEP was mostly low. Only the eastern part of the Vilkitsky Strait had an increase in sea ice concentration, with an abnormal value of about 10%.

It can be seen from Figure 10 that in July 2020, the sea ice concentration of the NEP of the Arctic was abnormally lower than the multi-year average, specifically in the East Siberian Sea and the Laptev Sea, where the anomalous value mostly remained below −40%.

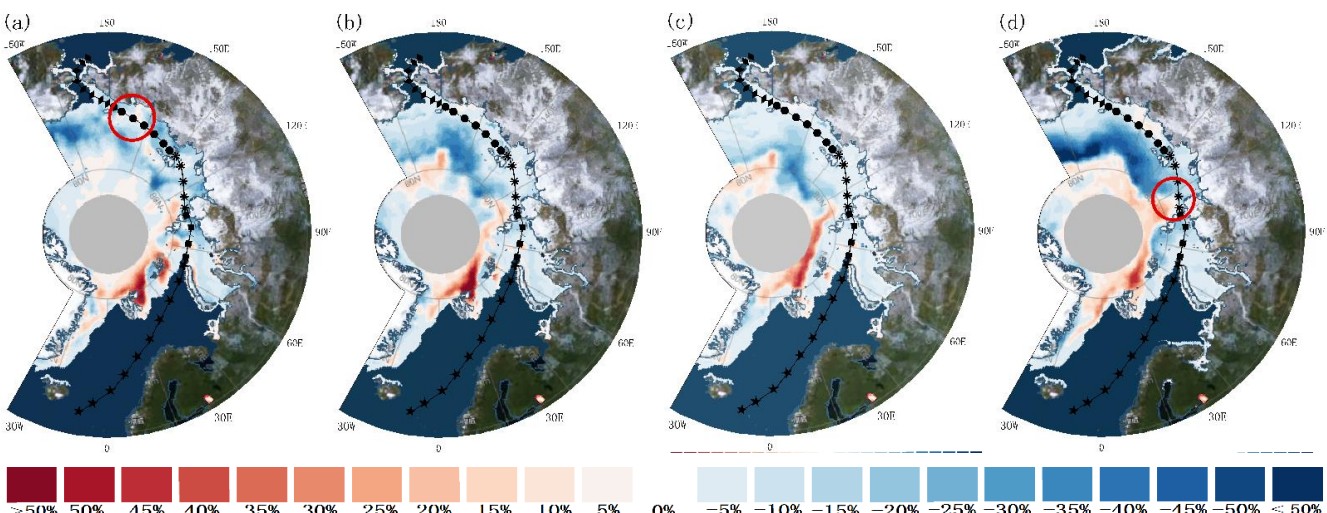

**Figure 9.** The anomalous distribution of sea ice concentration from July to October 2019 (**a–d**). (The outliers are based on the period from 2011 to 2020, the red circles represent the special areas).

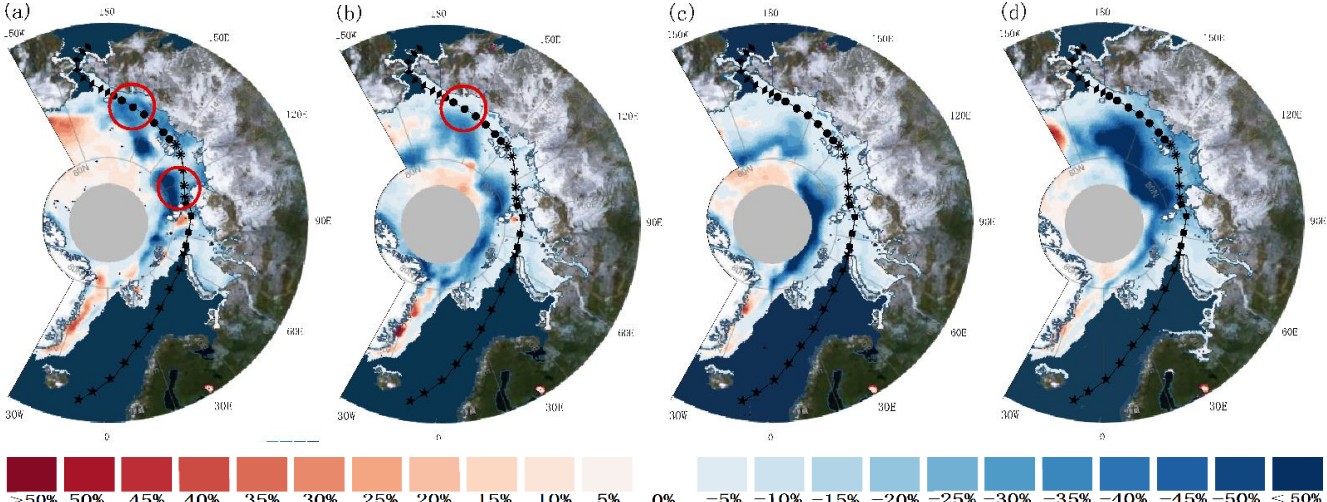

**Figure 10.** The anomalous distribution of sea ice concentration from July to October 2020 (**a–d**). (The outliers are based on the period from 2011 to 2020, the red circles represent the special areas).

In August, the sea ice concentration anomalies in most sea areas of the NEP were mostly between −20% and 0, and only in the central part of the East Siberian Sea, the anomalies were around −40%.

In September, the sea ice of the NEP changed a little.

In October, the sea ice concentration of the NEP was still abnormally lower than the multi-year average, and the abnormal value was far less than 40%.

### 3.4. NEP Connectivity Analysis in Hot Years

According to the research of Shibata et al. [27], when the sea ice concentration is 10–30%, there is only a small amount of broken ice on the sea surface. At this time, the channel is very smooth. Therefore, we use the AMSR-E/AMSR2 sea ice concentration daily data from July to October in 2016, 2019 and 2020 to calculate the navigation window of the NEP with thresholds of 10% and 30%, respectively. Compared with the method of visual interpretation, the method of calculating NEP navigation conditions based on the daily average sea ice concentration is more objective and scientific [28].

It can be seen from Figure 11 that in the navigation window of the NEP with a sea ice concentration of 30% as the threshold, the opening time of the NEP in 2016 was from

mid-August to late October, lasting 72 days, and the opening time is longer. In 2016, the area where the opening time of the NEP was late was 2885.77 nautical miles, which was the Vilkitsky Strait where the ice condition of the NEP was the most severe. The area where the opening time ended earlier was 2098.74 nautical miles (the east side of the New Siberian Islands). In the navigation window of the NEP with a sea ice concentration of 10% as the threshold, the navigation time of the NEP in 2016 was from the end of August to the end of October, lasting 52 days, of which the area where the start and end time of the navigation was limited was 2361.09 nautical miles (the west side of the New Siberian Islands).

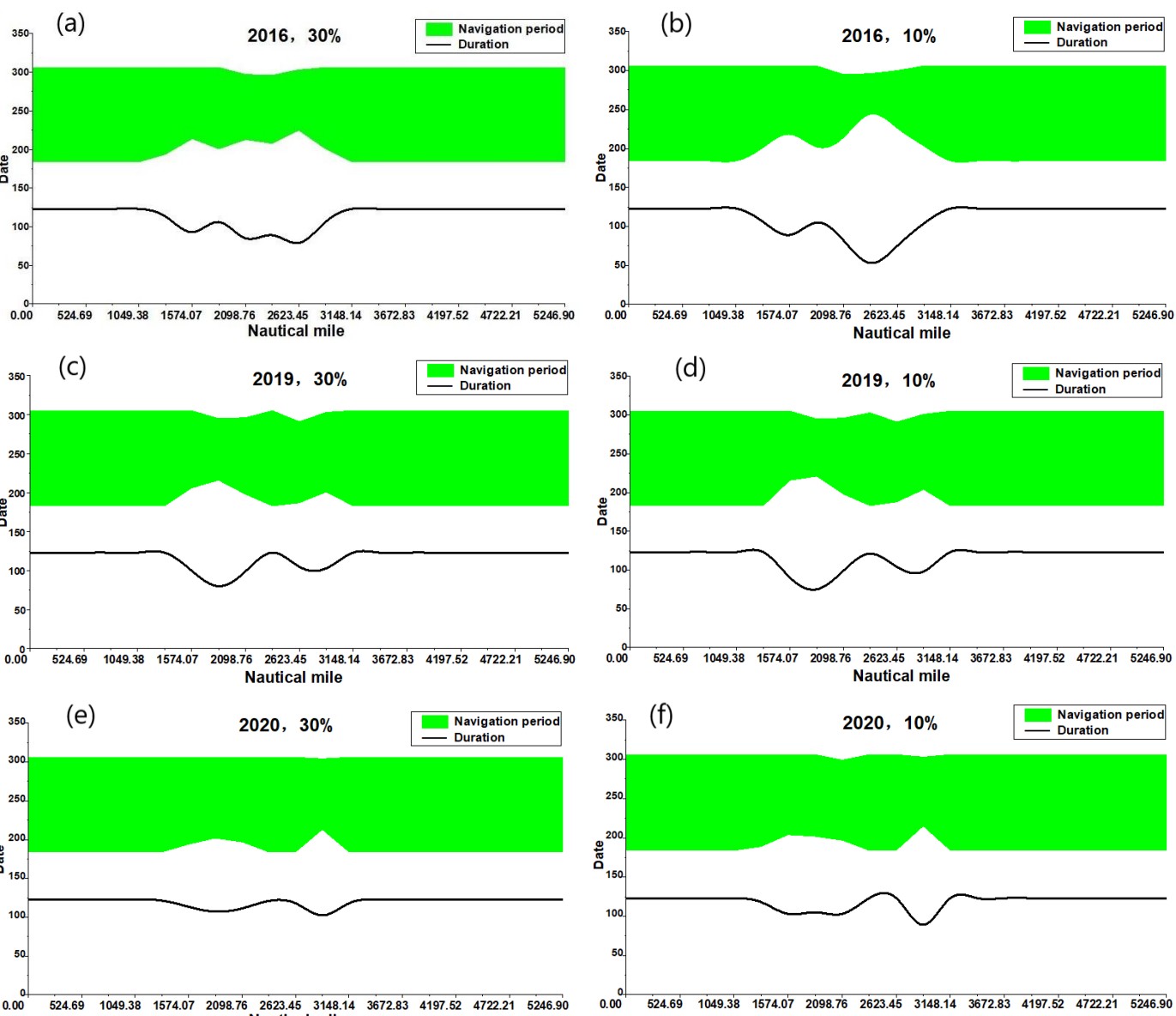

**Figure 11.** Navigation status of the Arctic NEP in 2016, 2019 and 2020. (The abscissa is the location of the NEP, and the ordinate is the date of the year in the NEP, the green area indicates the navigable period of the NEP in the current year, and the solid black line indicates the navigable duration of the navigable channel in the current year).

In 2019, in the navigation window of the NEP with a sea ice concentration of 30% as the threshold, the opening time of the NEP was from early August to mid-October, lasting 76 days. The area with the later opening time of the NEP was 1836.4 nautical miles (the central part of the East Siberian Sea), and the area where navigation ends earlier was 2623.43 nautical miles (Vilkitsky Strait). When the sea ice concentration was 10% as the

threshold, the navigation time of the NEP was from early August to mid-October, lasting 71 days. The area opened later was 1836.4 nautical miles (the central part of the East Siberian Sea), and the area ended earlier was 2623.43 nautical miles (Vilkitsky Strait).

In 2020, in the navigation window of the NEP with a sea ice concentration of 30% as the threshold, the opening time of the NEP was from the end of July to the end of October, lasting 92 days, becoming the longest year since the opening of the NEP. The areas that limited the start and end time of navigation were all at 2885.77 nautical miles (Vilkitsky Strait). Even in the navigation window with a sea ice concentration of 10% as the threshold, the opening period of the NEP in 2020 continued for 89 days, from the beginning of August to the end of October, and the restricted area of the NEP was 2885.77 nautical miles (Vilkitsky Strait). The navigation window obtained by using the 30% sea ice concentration as the threshold in this article was also verified in the research of Ji, M et al. [18]. At the same time, according to the International Shipping Network [29] about cargo ships sailing northward on the NEP, the time of independent sailing cargo ships sailing northward in 2016 and 2019 (as shown in Table 2) are both within the navigation window.

**Table 2.** Summary of cargo ship sailing northward.

| State | Year | Ship Name | Drive-In Time | Departure Time | Duration |
|---|---|---|---|---|---|
| | 2016 | Yongsheng | Nordkapp/20160913 | Bering Strait/20160922 | 9 |
| Independent sailing | 2019 | Tianxi | Nordkapp/20190826 | Bering Strait/20190905 | 10 |
| | | Tianyou | Bering Strait/20190905 | Nordkapp/20190916 | 11 |
| | | Tianhui | Bering Strait/20190909 | Nordkapp/20190920 | 11 |
| | | Daxiang | Nordkapp/20190906 | Bering Strait/20190916 | 10 |
| | | Tianqi | Nordkapp/20190912 | Bering Strait/20190923 | 11 |
| | | Tianen | Nordkapp/20190921 | Bering Strait/20191001 | 10 |
| | | Datai | Nordkapp/20190924 | Bering Strait/20191005 | 11 |

In the three high temperature years of 2016, 2019 and 2020, the navigation time of the NEP was more than 70 days when the threshold was 30%, and the navigation time was more than 50 days when the threshold was 10%. The navigation time was very long. The key navigable areas were the central waters of the East Siberian Sea at 1836.4 nautical miles, the New Siberian Islands at 2098.74–2361.09 nautical miles, and the Vilkitsky Strait at 2623.43–2885.77 nautical miles.

## 4. Analysis on the Spatial and Temporal Causes of Sea Ice Abnormality in Hot Years

Due to the time and space climatic factors of the Arctic Ocean having a certain lag in the melting and freezing of sea ice, here is the study of the changes in the time and space of sea ice formation in the summer of 2016, 2019 and 2020 from June to August, through sea level pressure, wind field, temperature, temperature anomalies, ice age and sea ice movement. Among them, the temperature anomaly is based on the monthly averages of June, July and August from 2011 to 2020.

### 4.1. 2016

It can be seen from Figure 12a that the pressure distribution in the Arctic region was uneven in June 2016. The central area of the Arctic was mainly low-pressure fields, while Greenland and Eurasia were high-pressure fields.

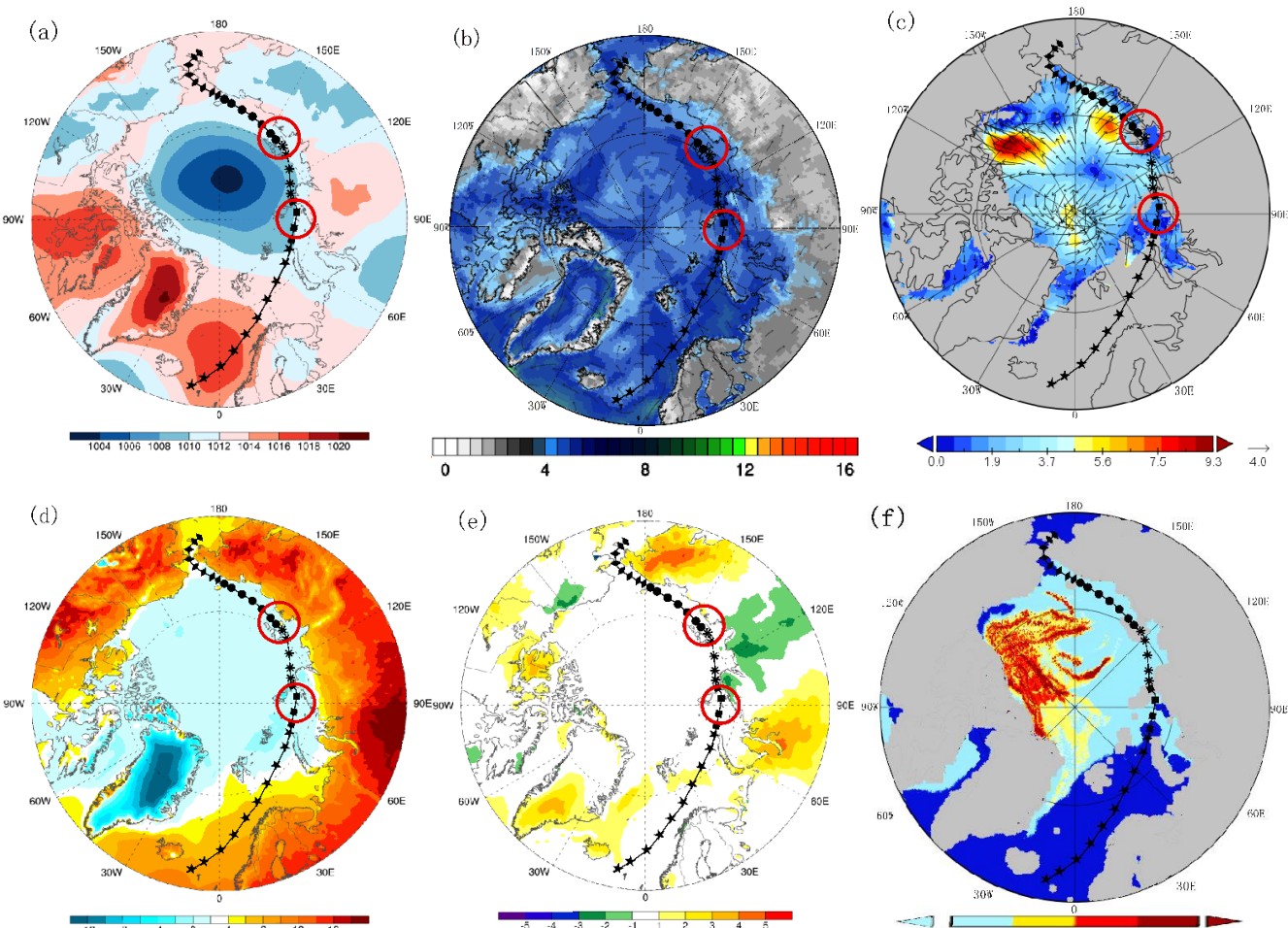

**Figure 12.** Distribution map of (**a**) sea level pressure (unit: hpa); (**b**) wind field (unit: m$^{-1}$); (**c**) sea ice movement (unit: m$^{-1}$); (**d**) temperature (unit: centigrade); (**e**) temperature anomaly (unit: centigrade) and (**f**) sea ice age (unit: year) in the Arctic summer in June 2016. (The red circles represent the special areas).

Under the influence of the pressure field, in June 2016, westerly winds prevailed in the East Siberian Sea, the Laptev Sea, and the Kara Sea. The Westerly wind also prevailed in the Vilkitsky Strait and the New Siberian Islands, while the northerly wind prevailed in the Barents Sea (Figure 12b).

The wind field promotes the sea ice movement. As shown in Figure 12c, in June 2016 the sea ice movement in the NEP was more apparent. Among them, in the New Siberian Islands, the sea ice movement was strong from northeast to southwest, but the sea ice movement on the west side of the Vilkitsky strait is not apparent.

It can be seen from Figure 12d,e that the temperature in the NEP in June 2016 was generally around 0 °C, while the temperature in the Barents Sea was higher, around 4 °C. Compared with the multi-year average temperature, the temperature on the NEP in June 2016 was not significantly abnormal.

According to Figure 12f, the sea ice in the East Siberian Sea, the Laptev Sea, and the Kara Sea is dominated by first year ice.

As shown in Figure 13a, in July 2016, there was only one high-pressure center near Greenland in the Arctic, and the other areas of the Arctic were all low-pressure fields.

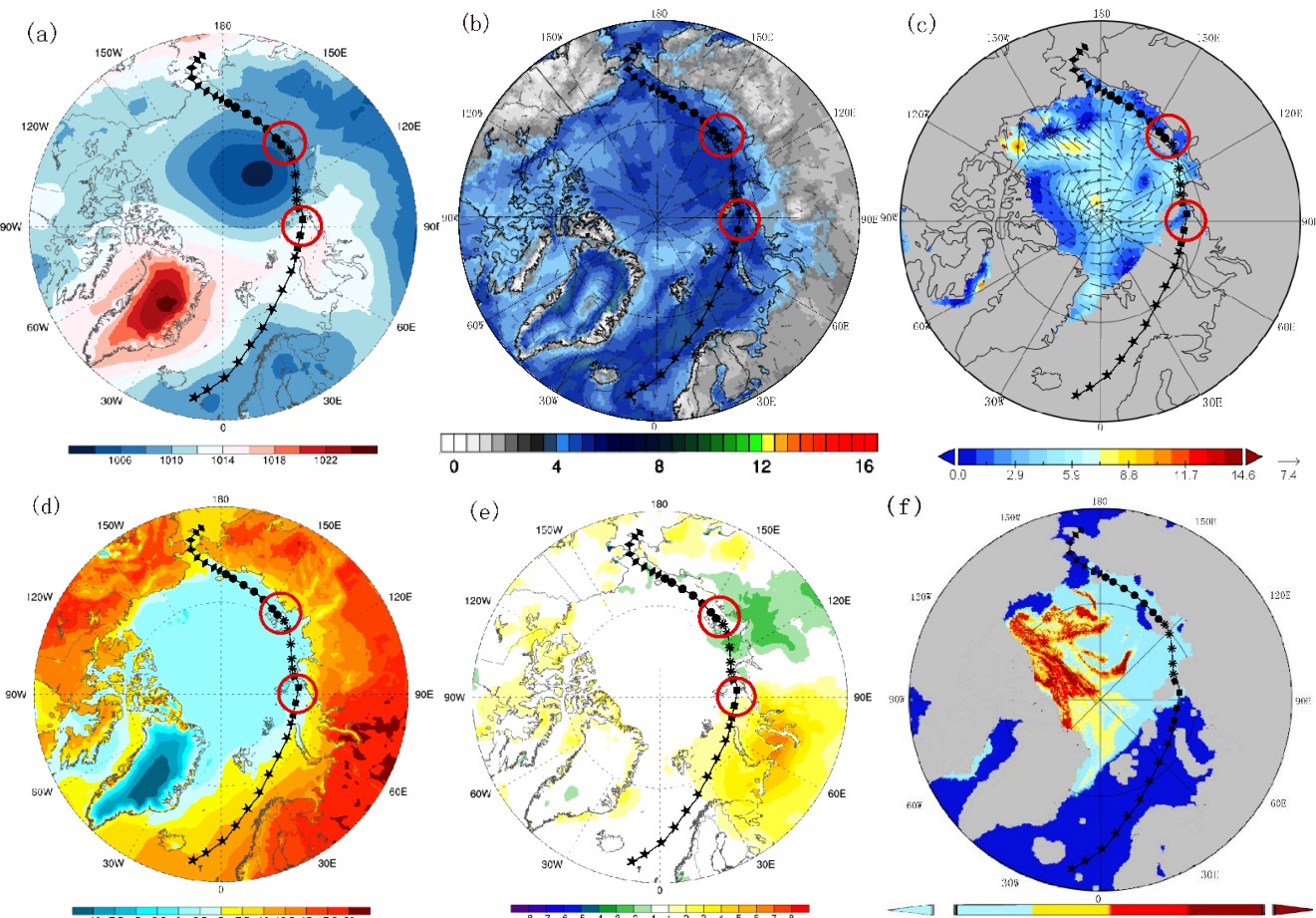

**Figure 13.** Distribution map of (**a**) sea level pressure (unit: hpa); (**b**) wind field (unit: m$^{-1}$); (**c**) sea ice movement (unit: m$^{-1}$); (**d**) temperature (unit: centigrades); (**e**) temperature anomaly (unit: centigrades) and (**f**) sea ice age (unit: year) in the Arctic summer in July 2016. (The red circles represent the special areas).

The East Siberian Sea and the New Siberian Islands were affected by the southwest wind, and the northwest wind prevailed in the Laptev Sea, the Kara Sea and the Vilkitsky Strait (Figure 13b).

It can be seen from Figure 13c that the sea ice movement on the New Siberian Islands and the west side of the Vilkitsky Strait in July 2016 was not strong, and the accumulation of sea ice was naturally tiny, which greatly promoted the opening of the NEP.

According to Figure 13d,e, the temperature in the NEP in July 2016 was relatively low, around −2.5 °C, while the temperature in the New Siberian Islands and the Vilkitsky Strait was not significantly abnormal compared to the multi-year average. According to Figure 13f, in July 2016, there was still more first year ice in the NEP.

It can be seen from Figure 14a that in August 2016, the center of the Arctic was low pressure, and the other areas were high-pressure fields.

In this pressure system, the East Siberian Sea, the Laptev Sea and the New Siberian Islands were all subject to southerly winds, while the Vilkitsky Strait, the Kara Sea and the Barents Sea were controlled by westerly winds, and the wind was relatively weak (Figure 14b).

It can be seen from Figure 14c that under the control of the wind field, in August 2016, sea ice movement near the NEP was not strong.

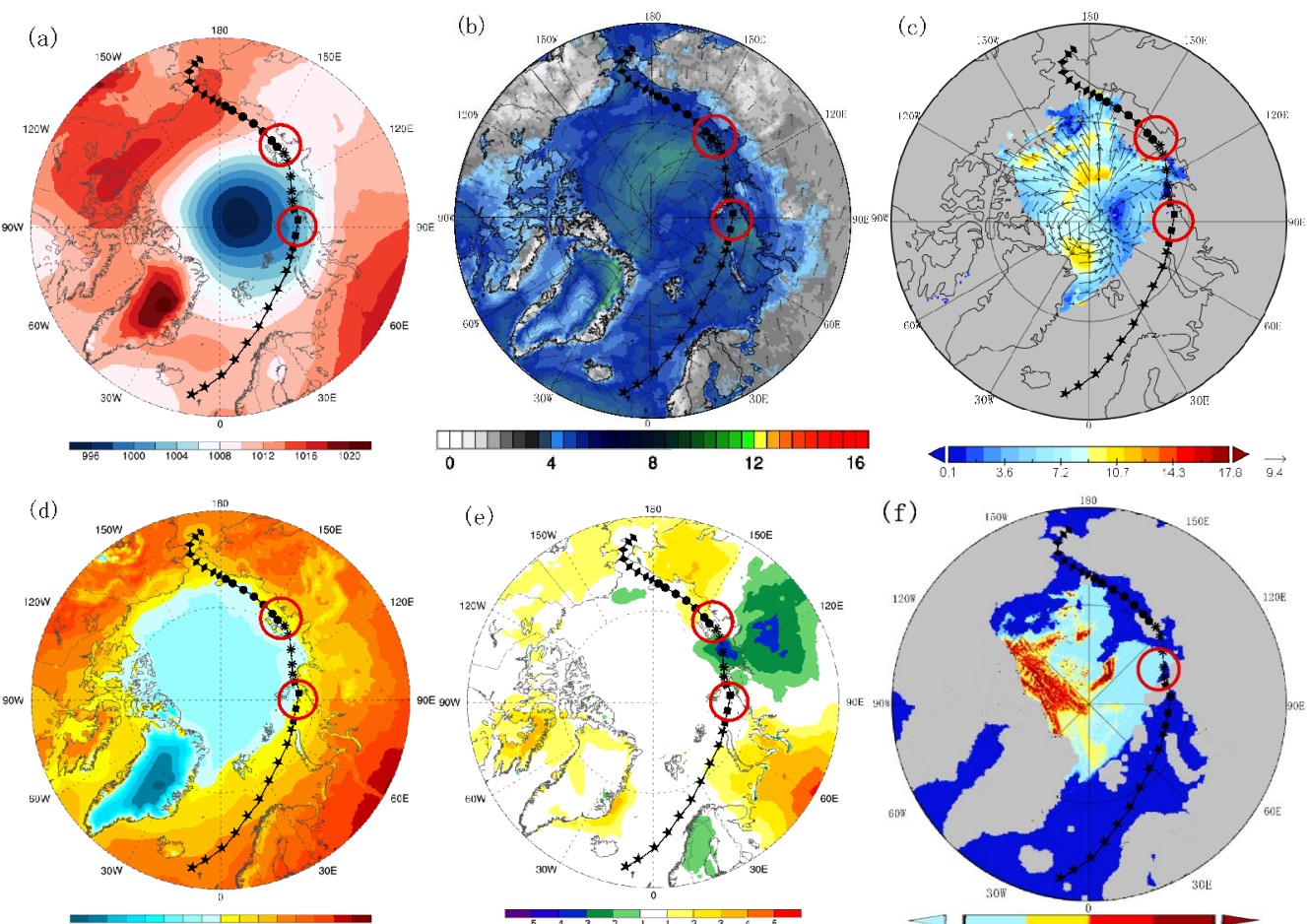

**Figure 14.** Distribution map of (**a**) sea level pressure (unit: hpa); (**b**) wind field (unit: m$^{-1}$); (**c**) sea ice movement (unit: m$^{-1}$); (**d**) temperature (unit: centigrades); (**e**) temperature anomaly (unit: centigrades) and (**f**) sea ice age (unit: year) in the Arctic summer in August 2016. (The red circles represent the special areas).

According to Figure 14d,e, the temperature in the NEP in August 2016 was relatively high, between 0–5 °C. In 2016, the two key areas for navigation, the New Siberian Islands and the Vilkitsky Strait, although there was no apparent abnormality compared to the multi-year average, the temperature remained between 0–5 °C, which greatly promoted the opening of the NEP.

In August 2016, the multi-year ice on the NEP was greatly reduced, and the Vilkitsky Strait were basically sea ice within one year (Figure 14f), which was easy to melt, promoted the opening of the NEP.

### 4.2. 2019

The navigable choke points of the NEP in 2019 were the central waters of East Siberian Sea and the east side of the Vilkitsky Strait.

It can be seen from Figure 15a that in June 2019, the Arctic and Eastern Hemisphere were dominated by low-pressure fields, and the Western Hemisphere was dominated by high-pressure fields.

Therefore, the East Siberian Sea, the Laptev Sea, the Kara Sea and the Vilkitsky Strait are affected by the southeast wind, and the Barents Sea is affected by the north wind (Figure 15b).

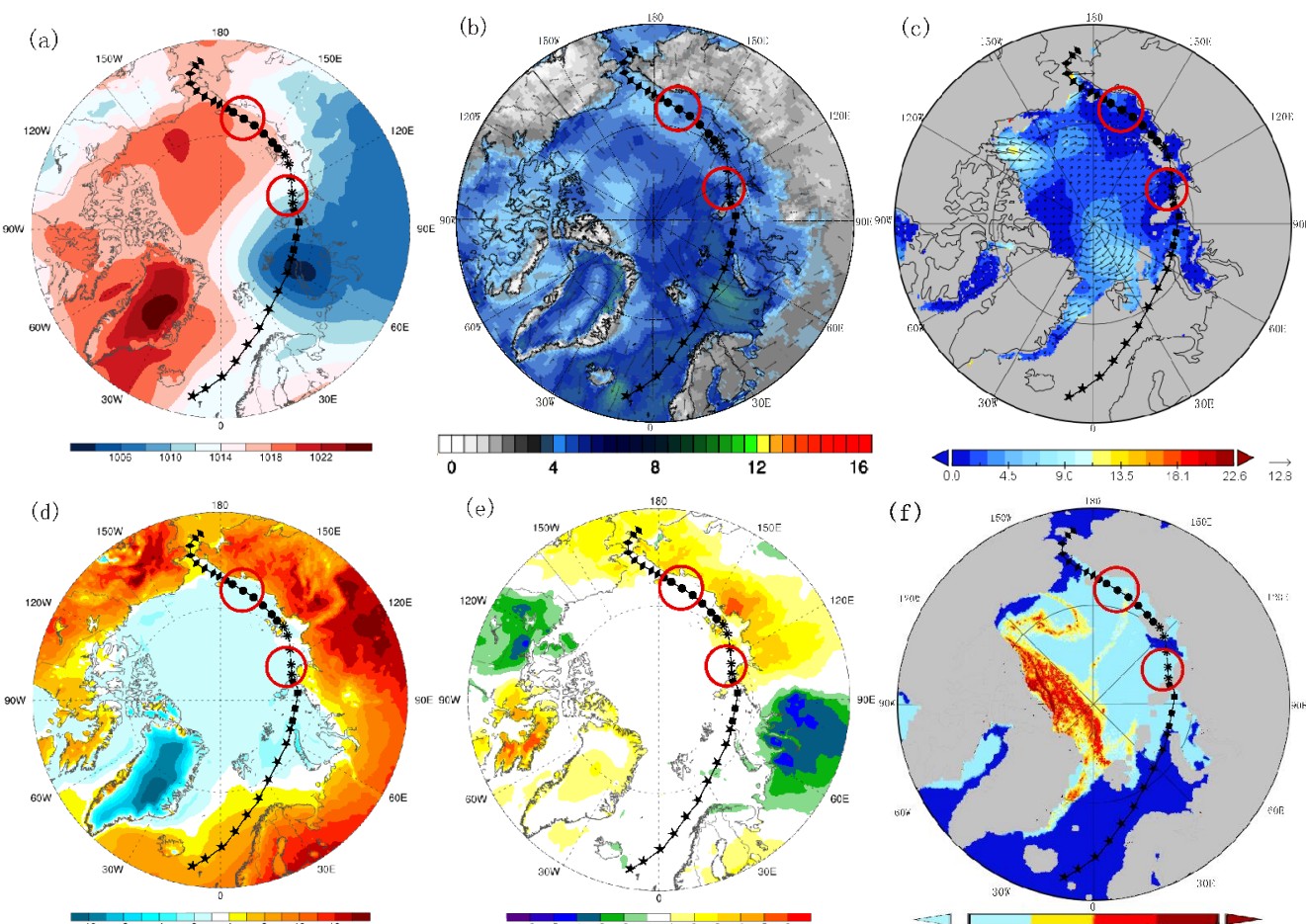

**Figure 15.** Distribution map of (**a**) sea level pressure (unit: hpa); (**b**) wind field (unit: m$^{-1}$); (**c**) sea ice movement (unit: m$^{-1}$); (**d**) temperature (unit: centigrades); (**e**) temperature anomaly (unit: centigrades) and (**f**) sea ice age (unit: year) in the Arctic summer in June 2019. (The red circles represent the special areas).

It can be seen from Figure 15c that in June 2019, although part of the sea ice movement occurred in the sea area near the NEP, it was not strong, which greatly promoted the opening of the NEP.

According to Figure 15d,e, the temperature in the NEP in June 2019 was between 0 and −2 °C, while the Barents Sea was warmer, between 0 and 8 °C. In the two key areas that will be open to navigation in 2019, the central waters of the East Siberian Sea and the Vilkitsky Strait, the annual average temperature is relatively high and there is no apparent abnormality.

In June 2019, the NEP was dominated by first year ice, and the central waters of the East Siberian Sea has a small amount of two-year ice.

It can be seen from Figure 16a that in July 2019, there were high-pressure centers in the central Arctic region and Greenland, and the remaining areas were low-pressure fields.

The East Siberian Sea, the Laptev Sea, the Kara Sea and the Vilkitsky Strait were all affected by the northerly wind. The Barents Sea was the easterly wind. The wind was generally weak (Figure 16b).

The sea ice movement in the waters near the NEP is not apparent (Figure 16c).

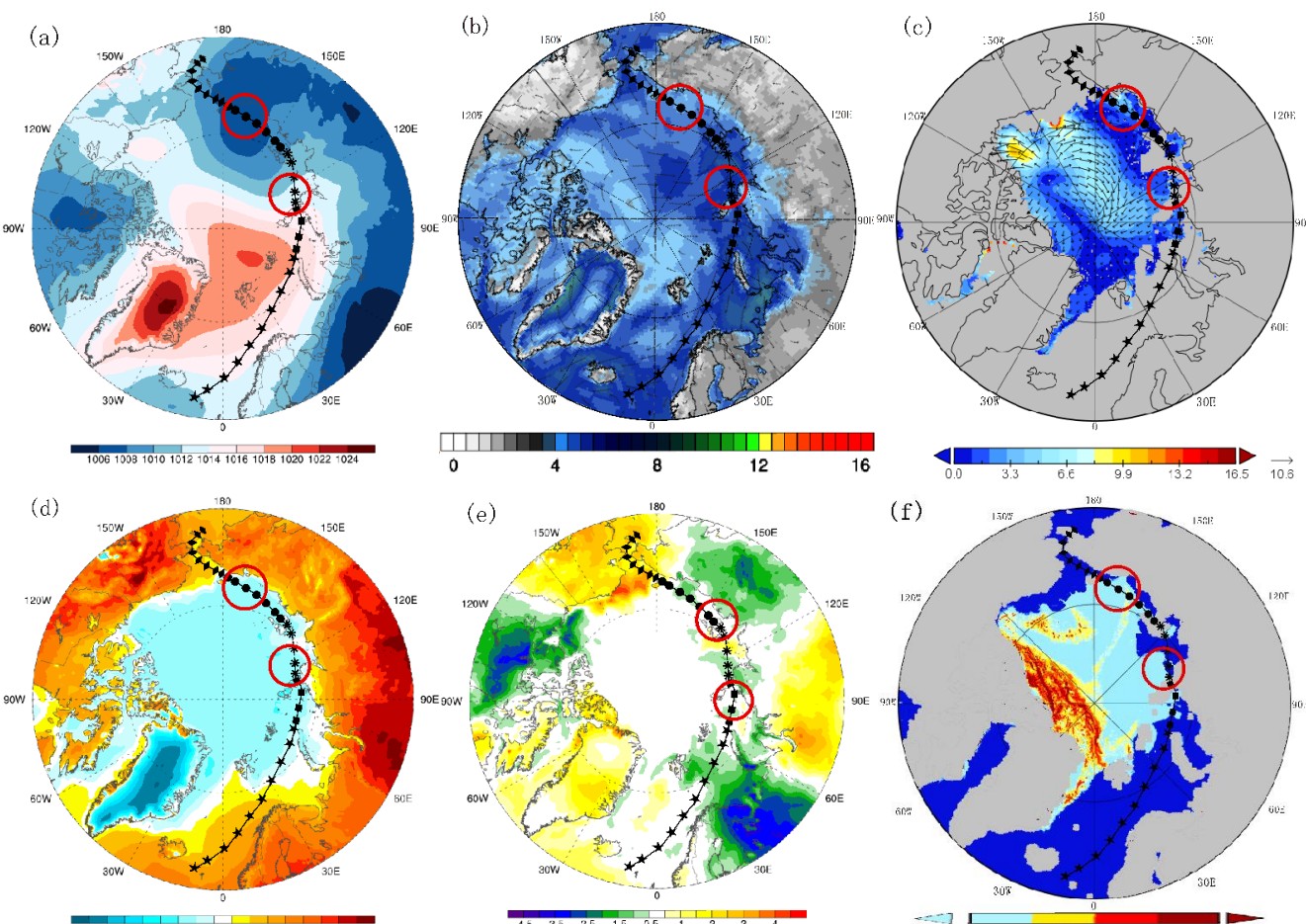

**Figure 16.** Distribution map of (**a**) sea level pressure (unit: hpa); (**b**) wind field (unit: m$^{-1}$); (**c**) sea ice movement (unit: m$^{-1}$); (**d**) temperature (unit: centigrades); (**e**) temperature anomaly (unit: centigrades) and (**f**) sea ice age (unit: year) in the Arctic summer in July 2019. (The red circles represent the special areas).

It can be seen from Figure 16d,e that the temperature in the NEP in July 2019 was basically between −4 and 2 °C, and the temperature in some sea areas showed negative anomalies, while the temperature in the central waters of East Siberian Sea and the Vilkitsky Strait did not show apparent anomalies.

According to Figure 16f, in July 2019, the central waters of the East Siberian Sea and the Vilkitsky Strait were still dominated by first year ice.

It can be seen from Figure 17a that in August 2019, only Greenland and the East Siberian Sea were high-pressure centers in the Arctic, and the rest were low-pressure fields.

Affected by the center of high and low pressure, the East Siberian Sea and Kara Sea were affected by westerly winds, and the Laptev Sea and Vilkitsky Strait were dominated by southerly winds (Figure 17b).

It can be seen from Figure 17c that in August 2019, only the central waters of the East Siberian Sean Sea and the east side of the Vilkitsky Strait experienced slight sea ice movement on the NEP.

According to Figure 17d,e, the temperature in the NEP in August 2019 was warmer, between 0–5 °C, and there was no apparent abnormality in most sea areas. The temperature in the Laptev Sea was positive compared to the multi-year average. In August, the temperature reached 5–10 °C, which greatly promoted the opening of the NEP. In August 2019, the NEP was almost free of sea ice.

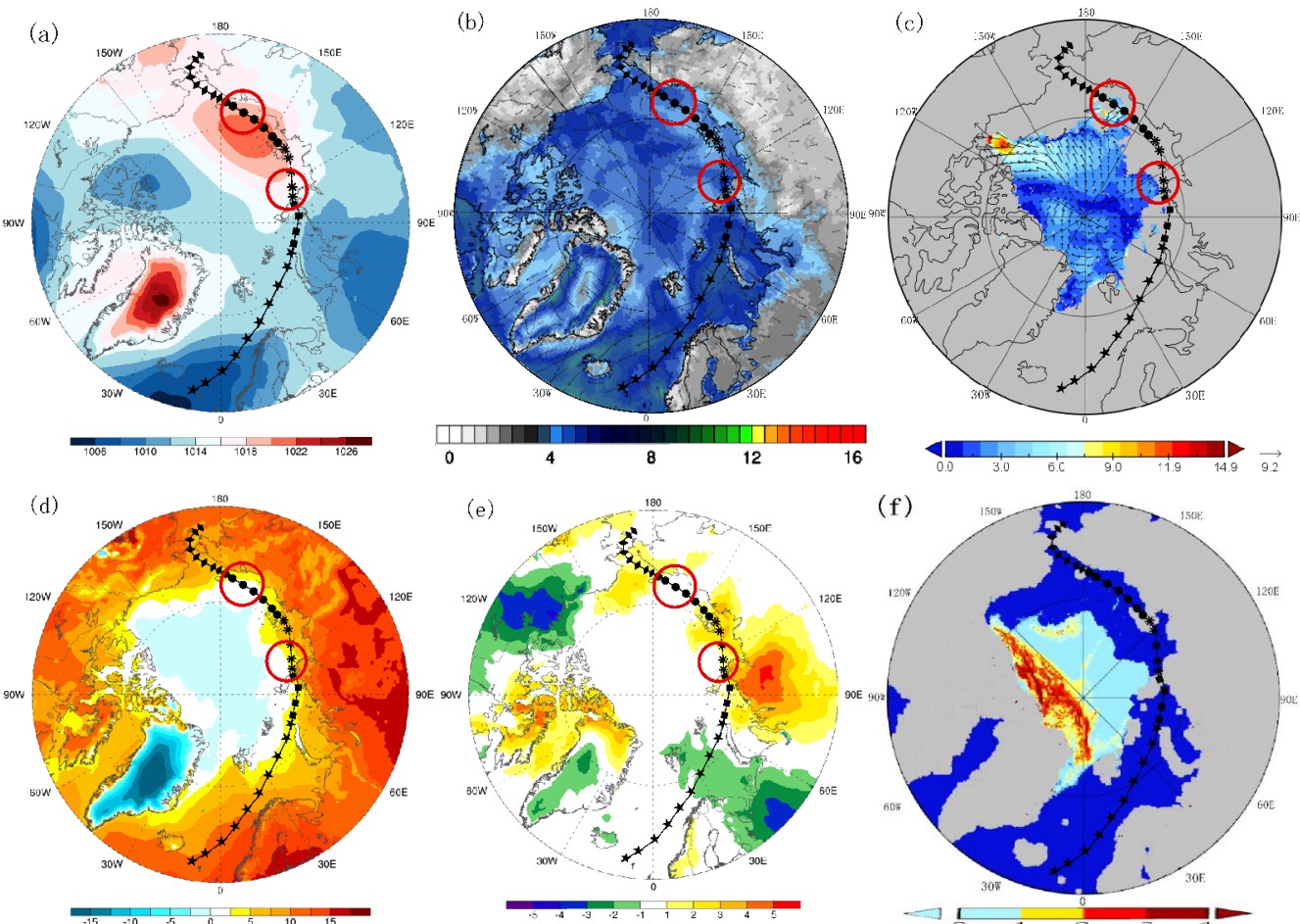

**Figure 17.** Distribution map of (**a**) sea level pressure (unit: hpa); (**b**) wind field (unit: m$^{-1}$); (**c**) sea ice movement (unit: m$^{-1}$); (**d**) temperature (unit: centigrades); (**e**) temperature anomaly (unit: centigrades) and (**f**) sea ice age (unit: year) in the Arctic summer in August 2019. (The red circles represent the special areas).

*4.3. 2020*

In 2020, the navigable choke points of the NEP were the New Siberian Islands and the west side of the Vilkitsky Strait.

It can be seen from Figure 18a that in June 2020, in the Arctic, the areas near the Canadian Islands, Greenland and the Barents Sea were high-pressure fields, and the Laptev Sea, Kara Sea and Eurasia were low-pressure fields.

The East Siberian Sea, the Kara Sea and the Barents Sea were affected by the northerly winds, the Laptev Sea and the New Siberian Islands were affected by the southerly winds, and the Westerly wind prevailed in the Vilkitsky Strait (Figure 18b).

It can be seen from Figure 18c that in June 2020, the sea ice movement in the NEP was strong, but there was less sea ice movement in the New Siberian Islands and the west side of the Vilkitsky Strait.

According to Figure 18d,e, the temperature in the NEP in June 2020 was mostly between 0 and 2.5 °C. Compared with the multi-year average, there was mostly no abnormal change in temperature.

In the East Siberian Sea, the New Siberian Islands and the west side of the Vilkitsky Strait, there was a large amount of first year ice (Figure 18f).



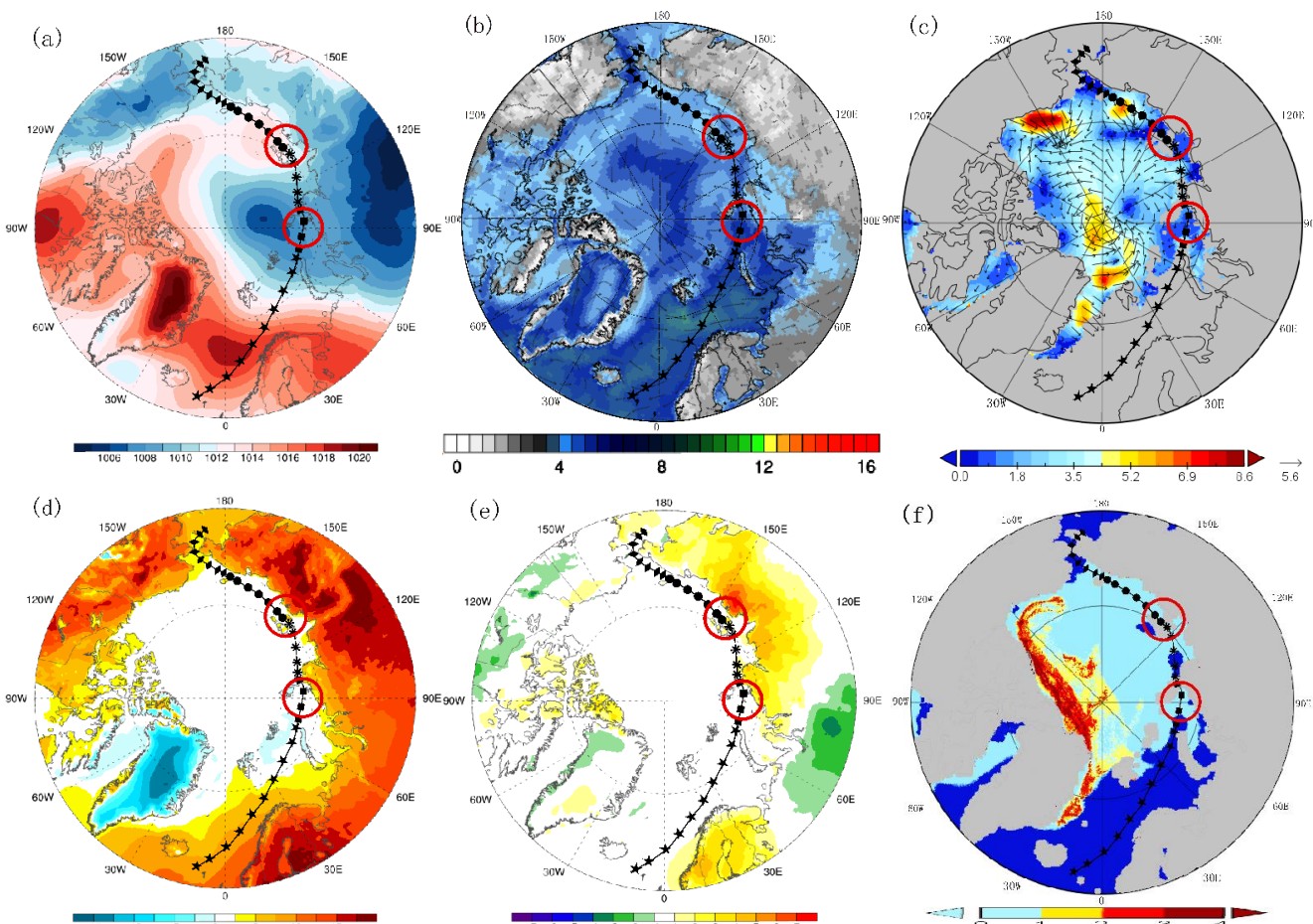

**Figure 18.** Distribution map of (**a**) sea level pressure (unit: hpa); (**b**) wind field (unit: m$^{-1}$); (**c**) sea ice movement (unit: m$^{-1}$); (**d**) temperature (unit: centigrades); (**e**) temperature anomaly (unit: centigrades) and (**f**) sea ice age (unit: year) in the Arctic summer in June 2020. (The red circles represent the special areas).

It can be seen from Figure 19a that in July 2020, there were high-pressure centers in the central Arctic and Greenland, and the rest were low-pressure fields.

Affected by the center of high and low pressure, northeasterly wind prevailed in the East Siberian Sea, Laptev Sea, and Kara Sea. The New Siberian Islands and the Vilkitsky Strait were affected by the east wind, and in the Barents Sea the southeast wind prevailed (Figure 19b).

It can be seen from Figure 19c that, affected by the wind field, in July 2020 sea ice near the NEP moved away from the NEP, and it was not easy for it to accumulate on the coast of the NEP, which was very conducive to the opening of the NEP.

According to Figure 19d,e, the temperature in the NEP in July 2020 was between −2.5 and 2.5 °C. Among them, the two navigable choke points of the NEP in 2020, the New Siberian Islands and the west side of the Vilkitsky Strait, the temperature was between 0–2.5 °C. Compared with the multi-year average, the temperature in the two key areas was a positive anomaly, the abnormally high was about 1.5 °C, which greatly promoted the opening of the NEP.

It can be seen from Figure 19f that in July 2020 a small amount of first year ice was distributed in the waters of the East Siberian Sea, the New Siberian Islands, and the west side of the Vilkitsky Strait.

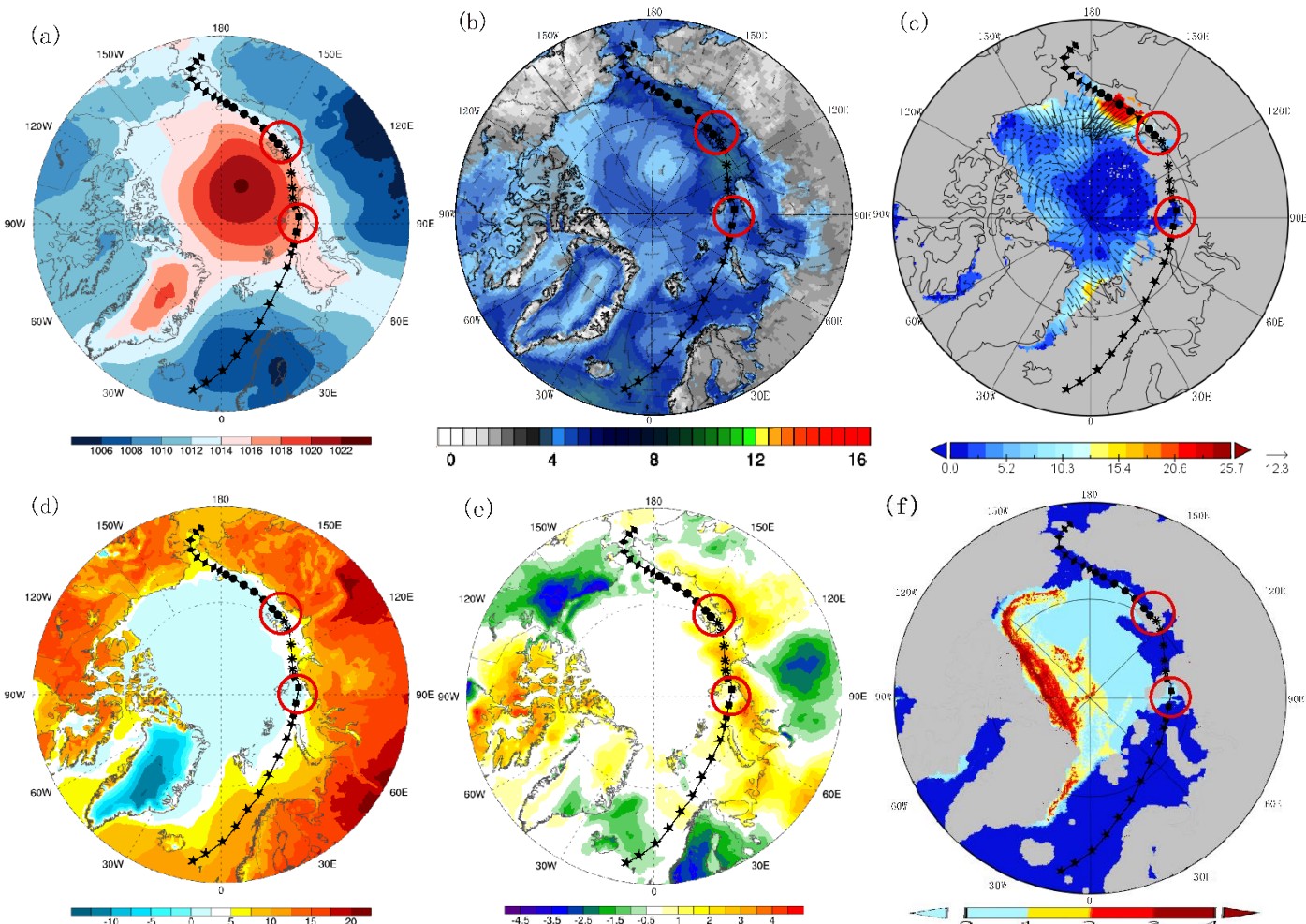

**Figure 19.** Distribution map of (**a**) sea level pressure (unit: hpa); (**b**) wind field (unit: m$^{-1}$); (**c**) sea ice movement (unit: m$^{-1}$); (**d**) temperature (unit: centigrades); (**e**) temperature anomaly (unit: centigrades) and (**f**) sea ice age (unit: year) in the Arctic summer in July 2020. (The red circles represent the special areas).

As shown in Figure 20a, in August 2020, there were high-pressure centers in the Arctic, Laptev Sea and Greenland, and low-pressure fields in the rest of the area.

Under the action of this pressure field, the NEP also formed different wind field effects (Figure 20b). The East Siberian Sea and the New Siberian Islands were affected by the northeasterly wind, while the Laptev Sea, Kara Sea and Barents Sea were dominated by warm and humid south wind from over Eurasia. The wind in the NEP was relatively weak.

It can be seen from Figure 20c that in August 2020 only slight sea ice movement occurred on the west side of the Vilkitsky Strait on the NEP, and the rest of the sea area had no sea ice movement, which was very conducive to the opening of the NEP.

According to Figure 20d,e, in August 2020 the temperature in the NEP was higher, all above 0 °C, and the temperature in the Laptev Sea, Kara Sea and Barents Sea even reached 5 °C. Compared to the multi-year average, the temperature in the Laptev Sea and Kara Sea in August 2020 was unusually high. This was specifically the case in the Vilkitsky Strait, the most dangerous area for navigation on the NEP. The temperature was higher than the average of the previous ten years. The value was nearly 3 °C higher, which greatly promoted the ablation of the sea ice in the strait, thereby prolonging the navigation time of the NEP.

According to Figure 20f, there was no sea ice in the NEP in August 2020 and the sea was open, which greatly promoted the opening of the NEP.

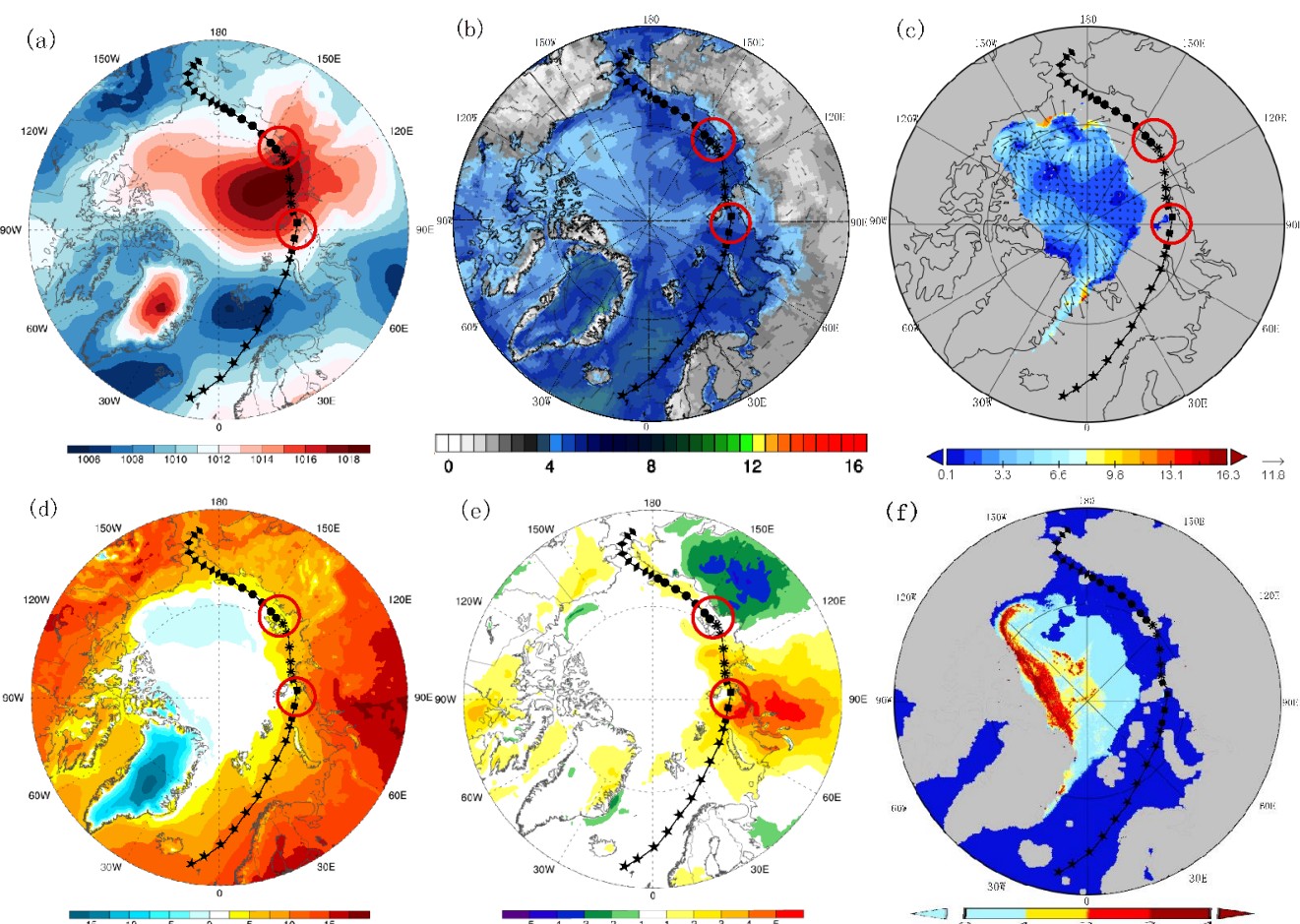

**Figure 20.** Distribution map of (**a**) sea level pressure (unit: hpa); (**b**) wind field (unit: m$^{-1}$); (**c**) sea ice movement (unit: m$^{-1}$); (**d**) temperature (unit: centigrades); (**e**) temperature anomaly (unit: centigrades) and (**f**) sea ice age (unit: year) in the Arctic summer in August 2020. (The red circles represent the special areas).

### 4.4. Discussion

Because of the abnormally high temperature in these three years, the opening time of the NEP was relatively long. The opening time of the NEP in 2016 was mainly restricted to the New Siberian Islands. The sea ice here was susceptible to the influence of the sea level pressure field and causes cyclonic motion. The sea ice was pushed to accumulate here, causing the sea ice concentration here to be higher than other areas along the NEP. The opening throat area of the NEP in 2019 was the Vilkitsky Strait and the central waters of the East Siberian Sea. The Vilkitsky Strait was the most critical area for the opening of the NEP. This was due to the center of low pressure in the Arctic summer, which produced a strong wind field, which promoted the movement of sea ice in the Arctic Ocean to the Vilkitsky Strait. The sea ice gathered on the east side of the Severnaya Zemlya, resulting in complicated sea ice conditions. In the middle of the East Siberian Sea, there was often a certain amount of multi-year ice in winter and spring, and the ablation of multi-year ice was slow, which also affected the navigation of the NEP in summer. The opening throat area of the NEP in 2020 was also the Vilkitsky Strait.

Although there were key navigable areas in the NEP in 2016, 2019 and 2020, the ice conditions in the throat areas were pretty good, which promoted the opening of the NEP in these three high-temperature years. The main driving factors for the opening of the NEP in 2016 were the low wind power, weak accumulation of sea ice movement, and the ice age was dominated by 1–2 years of ice, which was easy to melt, which greatly promoted the opening of the NEP in 2016. The main factors contributing to the navigation of the NEP in 2019 were the weak wind, low sea ice movement and high temperature, which were likely

to promote the ablation of sea ice, which were conducive to the opening of the NEP. The opening of the NEP took the longest time in 2020. Similar to 2016, the main driving factors for the navigation of the NEP in 2020 were the short ice age, which was dominated by ice for 1–2 years. In addition, the sea ice movement of the NEP had little effect in 2020. The key navigable areas, the New Siberian Islands and the west side of the Vilkitsky Strait basically had no sea ice movement, so it was naturally difficult for ice to accumulate and block the entrance. Compared with 2016, the temperature of the NEP was unusually higher in 2020, which promoted the navigation of the NEP.

## 5. Conclusions

(1) The sea ice extent of 2016, 2019 and 2020 all reached the minimum sea ice extent of that year in mid-September. Among them, the sea ice extent in September 2020 was the smallest, $3.74 \times 10^6$ km$^2$. The sea ice extent in September 2016 and 2019 was similar. The sea ice extent in 2016 was $4.17 \times 10^6$ km$^2$, and in 2019 it was $4.2 \times 10^6$ km$^2$.

(2) Compared with the multi-year average sea ice extent, the monthly sea ice extent in the summer (July-October) in 2016, 2019 and 2020 were all lower than the average level, and the sea ice extent in October has the largest difference. The difference in October 2016 was $4.9 \times 10^5$ km$^2$, the difference in October 2019 was $9.1 \times 10^5$ km$^2$, and the difference in October 2020 was $1.36 \times 10^6$ km$^2$. Compared to the multi-year average, the summer of 2020 (July-October) in these three years had the largest change of sea ice extent, followed by 2016, the change trend in 2019 was relatively small.

(3) The ice condition of the NEP in 2016 was relatively good. In July, the sea ice concentration of the East Siberian Sea and the Laptev Sea was about 60%, and the sea ice concentration of these two sea areas was abnormally higher than the multi-year average by about 30%. From August to October, the sea ice concentration in the sea area passed by the NEP was below 30%, which was little hindrance to navigation.

(4) The ice conditions in 2019 were better than those in 2016. In July, 30–60% of the sea ice was distributed in the Vilkitsky Strait and the East Siberian Sea in the NEP. The sea ice concentration in the middle of the East Siberian Sea was 10% higher than the multi-year average, and navigation was more difficult. In August and September there was an ice-free area in the NEP. In October, only in the eastern part of the Vilkitsky Strait was distributed a small amount of about 30% of the sea ice in the NEP.

(5) The NEP had the best ice conditions in 2020. In July, only a small amount of about 50% of the sea ice was distributed in the western part of the Vilkitsky Strait. The ice conditions in the other segments were great. The sea ice concentration along the NEP was abnormally lower than the multi-year average, which greatly promoted the opening of the NEP. From August to October, there was very little sea ice in the NEP, and the sea ice concentration was still abnormally lower than the multi-year average by 40%.

(6) When the sea ice concentration was 30% and 10%, respectively, as the threshold, the opening time of the NEP in 2016 was from mid-August to late October and from the end of August to late October. The navigable throat area was the New Siberian Islands. The opening time of the NEP in 2019 was from early August to mid-October and from early August to mid-October. The navigable throat areas were the Vilkitsky Strait and the central waters of the East Siberian Sea. 2020 was the longest year since the opening of the NEP. The opening times were from the end of July to the end of October and from the beginning of August to the end of October. The navigable throat area was the Vilkitsky Strait.

**Author Contributions:** Conceptualization, G.L.; Data curation, G.L. and Y.L.; Investigation, F.J. and M.J.; Methodology, G.L. and M.J.; Supervision, Y.L. and Y.H.; Validation, T.L.; Writing—original draft, G.L.; Writing—review and editing, M.J. All authors have read and agreed to the published version of the manuscript.

**Funding:** This research was funded by the National Natural Science Foundation of China (grant number 41471330); the Special Study on the Third Land Survey in Shandong Province (I) (grant number Y220004202000004_001); the National Science Fund subsidized project (grant number 41976184); and the Major scientific and technological innovation projects in Shandong Province (grant number 2019JZZY020103). The APC was funded by the National Natural Science Foundation of China (grant number 41471330).

**Institutional Review Board Statement:** Not applicable.

**Informed Consent Statement:** Not applicable.

**Data Availability Statement:** The sea ice concentration data comes from the AMSR-E/AMSR2 high-resolution daily sea ice concentration data provided by the Institute of Environmental Physics (IUP) of the University of Bremen: https://seaice.uni-bremen.de/start/, accessed on 18 October 2021. The sea ice extent data comes from the passive microwave data from the Defense Meteorological Satellite Program (DMSP) F17 and F18 Special Sensor Microwave Imager/Sound Device (SSMIS): https://nsidc.org/data/NSIDC-0192/versions/3, accessed on 18 October 2021. The sea level pressure, 2 m of near-surface air temperature and temperature anomalies come from NOAA, ESRL: https://psl.noaa.gov/repository/model/compare, accessed on 18 October 2021. The wind field data comes from the University of Maine: https://climatereanalyzer.org/reanalysis/monthly_maps/, accessed on 18 October 2021. The sea ice age comes from NSIDC: https://nsidc.org/data/NSIDC-0611/versions/4, accessed on 18 October 2021. The sea ice motion data comes from NSIDC: https://nsidc.org/data/NSIDC-0116/versions/4, accessed on 18 October 2021.

**Conflicts of Interest:** The authors declare no conflict of interest.

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
