# Peer review of "Analysis of the Spatial and Temporal Variation of Sea Ice and Connectivity in the NEP of the Arctic in Summer in Hot Years"

_jmse, doi:10.3390/jmse9111177_

Round 1

Reviewer 1 Report

Jmse-1173558- review

Guochong Liu, Min Ji, Fengxiang Jin, Ying Li, Yawen He and Ting Li: Analysis of the Spatial and Temporal Variation of Sea Ice and Connectivity in the NEP of the Arctic in Summer in Hot Years

This manuscript presents investigated sea ice conditions in the Arctic during three summers (2016, 2019 and 2020) with unusually long periods for ship traffic through the Northeast Passage. Stated in the introduction, they also aim at summarizing causes for these long open periods from data on air pressure, wind, temperature and se ice drift. However, their analysis of causes is weak. The observations are nicely presented in figures. The text, however is largely just describing the figures with very little discussion of connections between ice conditions and causes. I think your message drowns in repeated details of what you see in the figures. The information is repeated three times (2nd in summary and 3rd in abstract), which makes it boring to read. Some suggestions are given below.

Abstract

I thought it was not common to use abbreviations in the abstract. There are too many details in the abstract. Not necessary to repeat everything.

General through the text of chapters 1 and 2: the are many cases where words are split even though the letters in the word fits in the same line. Find examples in lines 67, 68, 80, 86, 87, 96, 111, 128

  1. Introduction

Line 42: write Northeast Passage (NEP)

Line 95: here you have an objective to ‘summarize the causes of sea ice anomalies in the NEP in the past three years’. You probably mean the three selected warm years. Moreover, the reader expects to see a discussion around this later in the manuscript, which in reality is very brief.

Line 97: ‘law of sea ice changes’? You probably mean  ‘causes of sea ice changes’.

  1. Research area, data and method

Line 109: M. Ji is first author of reference 18

Line 114: Table 2 is not in the manuscript.

  1. Analysis of the NEP Ice Condition and Connectivity in Hot Years

Lines 216-219: these formulae are unnecessary to write in the text, as they are shown in Figure 4. Besides, the is an extra minus-sign in the equation for 2020.

Line 230: ‘red circle’, not ‘black circle’

Line 236: same as line 230

Line 288: I believe it is 2016, not 2020.

Line 319: Not only 2020 I presume.

Line 355: M. Ji is first author of reference 18

Line 359: Table 3 is not referred to in the text.

Lines 367-376: Where are the evidences for these statements? I don’t disagree with the content, but it comes too early in the manuscript. It should be placed in connection with a discussion around Figures 12-20.

  1. Analysis on the Spatial and Temporal Causes of Sea Ice Abnormality in Hot Years

Generally, the text describing the nine figure is a repetition of pretty much the same information, which becomes quite boring in the long run. It does not make it more exciting to read when there is no information about where New Siberian Islands and Vilkitsky Strait are on the map. Most of the time the authors do not reveal any understanding of the relationship between air pressure field and wind field (except e.g. lines 430-432). And they never connect ice drift to the wind field, although this is argued to be important in the last paragraph of chapter 3. I do sense a tendency for the ice to be drifting at an angle to the right of the wind, in accordance with Ekman-theory.

I think you can do better than this, and really discuss the importance of wind and ice drift, compared to temperature and ice conditions.

Line 521: Sea ice younger than 1 year is generally called first year ice, not one-year ice.

Line 569: I suggest you call this section 4.4 Discussion in stead of Summary, and merge it with the last paragraph of Section 3.

  1. Conclusion

This section is more like a summary than conclusion, where you repeat text you have earlier in the manuscript (copy paste, including the extra minus in equation for 2020 in Figure 4). I suggest you shorten points 1-7, and rather keep point 8 adjusted to the new Discussion in section 4.4.

Figures

                Figure 2: Add these names in the map, so the reader unfamiliar to the Arctic region can understand where they are: Vilkitsky Strait, Severnaya Zemlya, New Siberian Islands, Greenland, Canadian Islands (or Archipelago), Eurasia.

                Figure 3: The yellowish colors are very hard to see. Perhaps use a gray scale on other years than 2016, 2019 and 2020. Except 2012, this minimum ice year should be possible to see in the figure.

                Figures 8-11: Use bigger fonts. It is impossible to see which color is positive and which is negative (I assume re is positive, i.e higher concentration than normal)

Figure 11: Bigger fonts.

Figures 12-20: Add units (mbar, m/s, cm/s, centigrades, year?). Very hard to see wind arrows in b. Font in f is too small.

Author Response

Please see the coverletter for details. 

Reviewer 2 Report

  1. Decipher the abbreviation NEP in the main text.
  2. Decrypt the NSIDC.
  3. Decrypt the NWP.
  4. Decrypt AMSR-E/AMSR2
  5. In the Introduction section, make the reference to Figure 1.
  6. Table 2 is lost.
  7. Explain what the matlab means? Give a link to the description of the Matlab program.
  8. In text make the reference to Table 3.
  9. The text contains a large number of typos.

Author Response

Please see the coverletter for details. 

Round 2

Reviewer 1 Report

I think the changes you have made makes the paper more readable. The discussion is improved.